# Photoaging of Phenolic Secondary Organic Aerosol in the Aqueous Phase: Evolution of Chemical and Optical Properties and Effects of Oxidants

Wenqing Jiang[1,2], Christopher Niedek[1,2], Cort Anastasio[2,3], Qi Zhang[1,2*]

[1]Department of Environmental Toxicology, University of California, 1 Shields Ave., Davis, CA 95616, USA
[2]Agricultural and Environmental Chemistry Graduate Program, University of California, 1 Shields Ave., Davis, CA 95616, USA
[3]Department of Land, Air, and Water Resources, University of California, 1 Shields Ave., Davis, CA 95616, USA

*Correspondence to:* Qi Zhang (dkwzhang@ucdavis.edu)

**Abstract.** While gas-phase reactions are well established to have significant impacts on the mass concentration, chemical composition, and optical properties of secondary organic aerosol (SOA), the aqueous-phase aging of SOA remains poorly understood. In this study, we performed a series of long-duration photochemical aging experiments to investigate the evolution of the composition and light absorption of the aqueous SOA (aqSOA) from guaiacyl acetone (GA), a semivolatile phenolic carbonyl that is common in biomass burning smoke. The aqSOA was produced from reactions of GA with hydroxyl radical (•OH-aqSOA) or a triplet excited state of organic carbon ($^3$C*-aqSOA) and was then photoaged in water under conditions that simulate sunlight exposure in northern California for up to 48 hours. The effects of increasing aqueous-phase •OH or $^3$C* concentration on the photoaging of the aqSOA were also studied. High resolution aerosol mass spectrometry (HR-AMS) and UV-vis spectroscopy were utilized to characterize the composition and the light absorptivity of the aqSOA and to track their changes during aging.

Compared to •OH-aqSOA, the $^3$C*-aqSOA is produced more rapidly and shows less oxidation, a greater abundance of oligomers, and higher light absorption. Prolonged photoaging promotes fragmentation and the formation of more volatile and less light-absorbing products. More than half of the initial aqSOA mass is lost and substantial photobleaching occurs after 10.5 hours of prolonged aging under simulated sunlight illumination for $^3$C*-aqSOA and 48 hours for •OH-aqSOA. By performing positive matrix factorization (PMF) analysis of the combined HR-AMS and UV-vis spectral data, we resolved three generations of aqSOA with distinctly different chemical and optical properties. The first-generation aqSOA shows significant oligomer formation and enhanced light absorption at 340–400 nm. The second-generation aqSOA is enriched in functionalized GA species and has the highest mass absorption coefficients in 300–500 nm, while the third-generation aqSOA contains more fragmented products and is the least light-absorbing. These results suggest that intermediately-aged phenolic aqSOA is more light-absorbing than other generations, and that the light absorptivity of phenolic aqSOA results from a competition between brown carbon (BrC) formation and photobleaching, which is dependent on aging time. Although photoaging generally increases the oxidation of aqSOA, a slightly decreased O/C of the •OH-aqSOA is observed after 48 hours of prolonged

photoaging with additional •OH exposure. This is likely due to greater fragmentation and evaporation of highly oxidized compounds. Increased oxidant concentration accelerates the transformation of aqSOA and promotes the decay of BrC chromophores, leading to faster mass reduction and photobleaching. In addition, compared with •OH, photoaging by $^3$C* produces more low-volatility functionalized products, which counterbalances part of the aqSOA mass loss due to fragmentation and evaporation.

## 1. Introduction

Phenols, which are emitted from biomass burning (BB) through lignin pyrolysis (Schauer et al., 2001) and formed from the oxidation of aromatic hydrocarbons (Berndt and Böge, 2006), are important precursors for atmospheric secondary organic aerosol (SOA) and brown carbon (BrC) (Bruns et al., 2016; Mabato et al., 2022; Misovich et al., 2021; Smith et al., 2016; Sun et al., 2011; Yee et al., 2013). These compounds can form aqueous-phase SOA (aqSOA) at fast rates in atmospheric waters, through photoreactions with oxidants such as hydroxyl radical (•OH), excited triplet states of organic carbon ($^3$C*), and reactive nitrogen species (e.g., •NO, •NO$_2$, NO$^+$, and NO$_2^+$) (Jiang et al., 2021; Li et al., 2022b; Mabato et al., 2022, 2023; Pang et al., 2019; Yang et al., 2021; Yu et al., 2014). The mass yields of aqSOA from the phenolic precursors in atmospheric waters range from 50% to 140%, and the proposed formation pathways include oligomerization, functionalization (e.g., hydroxylation) and fragmentation (Arciva et al., 2022; Huang et al., 2018; Jiang et al., 2021; Ma et al., 2021; Smith et al., 2014, 2015, 2016; Sun et al., 2010; Yu et al., 2014, 2016). Reactions involving reactive nitrogen species can also lead to nitration and nitrophenol formation (Heal et al., 2007; Mabato et al., 2022; Pang et al., 2019; Yang et al., 2021). The resulting phenolic oligomers, multifunctional derivatives, and nitrophenols can absorb near-UV and visible light and contribute significantly to BrC formation in biomass burning emissions (Gilardoni et al., 2016; Li et al., 2022a; Misovich et al., 2021; Palm et al., 2020; Pang et al., 2019). In addition, humic-like substances (HULIS), which can induce oxidative stress and cause adverse health effects (Deng et al., 2022), are observed in phenolic aqSOA as well (Chang and Thompson, 2010).

Despite extensive research on the formation of aqSOA from phenols, the aging and degradation of phenolic SOA in water remain poorly characterized. Atmospheric lifetimes of SOA range from hours to weeks (Wagstrom and Pandis, 2009), during which chemical reactions can occur, leading to continuous aging and evolution of SOA. Functionalization (i.e., the addition of functional groups to the molecules) and fragmentation (i.e., the breaking of bonds within the molecules to form smaller species) are critical mechanisms in the aging of SOA that can greatly change the chemical composition and loading of aerosols (Kroll et al., 2009; Leresche et al., 2021; Shrivastava et al., 2017). Chemical aging can also influence the optical properties of SOA, as some reactions increase the light absorptivity while others cause photobleaching by destroying chromophores (Lee et al., 2014). Furthermore, fragmentation can result in the formation of volatile and semivolatile products, causing a loss of SOA mass and photobleaching (Kroll et al., 2015). Yu et al. (2016) studied the aqueous-phase photooxidation of phenol and methoxyphenols and observed that, as aging progresses, fragmentation reactions become increasingly dominant in comparison to oligomerization and functionalization reactions. However, a portion of the aqSOA appears to be resistant to fragmentation

and remains chemically unchanged even after prolonged exposure to simulated sunlight in the aqueous phase (Yu et al., 2016).

Similarly, in an environmental chamber study, O'Brien and Kroll (2019) reported that 70−90% of the α-pinene SOA mass remained in particles after an initial decay during photochemical aging.

The impacts of aging on the concentrations and properties of SOA in the atmosphere have been widely observed in biomass burning emissions (Brege et al., 2018; Chen et al., 2021; Garofalo et al., 2019; Kleinman et al., 2020; Zhou et al., 2017). For instance, aged wildfire plumes subjected to aqueous processing experience substantial losses in organic aerosol (OA) mass,

increases in SOA oxidation, and changes in optical properties (Che et al., 2022; Farley et al., 2022; Sedlacek et al., 2022). Aqueous-phase oxidation of organic molecules, including phenols, and the formation of SOA have been observed in residential wood burning smoke in both urban and rural environments as well (Brege et al., 2018; Kim et al., 2019; Stefania et al., 2016; Sun et al., 2010). In addition, in remote regions where aerosols are generally highly aged and have been subjected to more extensive aqueous-phase and heterogeneous processing, SOA is significantly more oxidized, less volatile, and more

hygroscopic compared to those in urban areas (Jimenez et al., 2009; Morgan et al., 2010; Ng et al., 2011; Zhang et al., 2011; Zhou et al., 2019).

Understanding the chemical aging process of SOA in the aqueous phase is important for better predicting the concentration of SOA in ambient air and assessing its potential impacts on climate and human health. Sunlight-triggered aqueous-phase reactions, such as direct photolysis of organics, nitrate, nitrite, and hydrogen peroxide, as well as energy and charge-transfer

reactions driven by $^3C^*$, significantly impact the chemical aging of SOA, leading to changes in particle composition and properties (Corral Arroyo et al., 2018; Ervens et al., 2011; Herrmann et al., 2015; Mabato et al., 2022). The extent of exposure of aqSOA to oxidants in atmospheric waters can vary widely, influenced by the concentration and residence time of oxidants. For example, the steady-state concentration of •OH can vary from $10^{-16}$ to $10^{-12}$ M (Herrmann et al., 2010) while that of $^3C^*$ can vary from $10^{-14}$ to $10^{-11}$ M (Kaur et al., 2019), depending on the solute concentration, which ranges from dilute fog/cloud

droplets to highly concentrated solutions in particle water. Exposure to elevated levels of oxidants can promote the formation of highly oxygenated SOA (Daumit et al., 2016; Kang et al., 2011; Lambe et al., 2015; Ng et al., 2010), but can also decrease SOA mass and facilitate a shift from the functionalization-dominant regime to the fragmentation-dominant regime (Lambe et al., 2012).

This study investigates the long-timescale aqueous aging of the aqSOA formed from the photooxidation of guaiacyl

acetone (GA). GA is a common component of biomass burning emissions and has been widely used as a model compound to study SOA formation in BB emissions. In previous work (Arciva et al., 2022; Jiang et al., 2021; Ma et al., 2021; Misovich et al., 2021; Smith et al., 2016), we examined the kinetics and mechanisms of aqSOA formation from the photoreactions of GA. Here, we extend this research to investigate the aqueous-phase photoreactions of GA with •OH and $^3C^*$ and propose reaction pathways of GA with •OH and $^3C^*$. The focus of our investigation is to study the impact of prolonged aqueous aging on the

chemical composition and optical properties of the aqSOA. Specifically, we examine the effects of •OH reaction and $^3C^*$ reactions induced by simulated sunlight for up to 72 hours and 14 hours, respectively, which correspond to approximately 21

days and 4 days of winter-solstice sunlight exposure in northern California (George et al., 2015). Furthermore, we examine the effects of light and additional oxidant exposure on the aging of the aqSOA.

## 2. Experimental Methods

### 2.1 Formation and Aging of Phenolic AqSOA

The initial reaction solution was prepared with 100 μM of guaiacyl acetone and either 100 μM of hydrogen peroxide ($H_2O_2$; as a source of •OH) or 5 μM of 3,4-dimethoxybenzaldehyde (3,4-DMB; as a source of $^3C^*$) in Milli-Q water. The pH of the solution was adjusted to 4.6 using sulfuric acid. These conditions were set to mimic wood burning-influenced cloud and fog waters (Jiang et al., 2021). The reaction solution was placed in a 400 mL Pyrex tube, continuously stirred and illuminated

inside a RPR-200 photoreactor system equipped with three different types of bulbs to roughly mimic sunlight (George et al., 2015). The steady-state concentration of •OH ([•OH]) is $2.6 \times 10^{-15}$ M in the •OH-mediated reaction, similar to the values observed in fog water (Kaur and Anastasio, 2017), and the [$^3C^*$] is $1.1 \times 10^{-13}$ M in the $^3C^*$-mediated reaction, about 2 times higher than in fog water (Kaur and Anastasio, 2018) (see Section 3.1 for more details). When ~95% of the initial GA has reacted (i.e., after 24 h of irradiation for the •OH reaction and 3.5 h for the $^3C^*$ reaction), the solution was separated into four

aliquots and moved into separate 110 mL Pyrex tubes for further aging. This aging occurred under four different conditions: 1) aging in the dark (tube wrapped with aluminum foil); 2) continued illumination without the addition of extra oxidant; 3) photoaging with the addition of 100 μM of $H_2O_2$; and 4) photoaging with the addition of 5 μM of 3,4-DMB. Small aliquots of the solutions were then periodically taken from each tube to measure the chemical composition and optical properties.

During the photoreaction, the solutions were continuously stirred. The Pyrex tubes were capped but not hermetically

sealed, and the caps were briefly removed during sample collection. Due to the presence of oxygen in the reaction system, secondary reactive oxygen species (ROS) such as singlet oxygen ($^1O_2^*$), superoxide/hydroperoxyl radicals ($O_2•^-/HO_2•$) and •OH can be generated in the solution via energy transfer from $^3C^*$ to dissolved $O_2$ (Vione et al., 2014; Zepp et al., 1977), electron transfer from the intramolecular charge-transfer complex of DMB ($DMB•^{+/•-}$) to $O_2$ (Dalrymple et al., 2010; Li et al., 2022b), and the reactions between DMB ketyl radical and $O_2$ (Anastasio et al., 1997). However, according to our previous

studies (Smith et al., 2014), singlet oxygen is expected to contribute only minimally to the oxidation of GA in this reaction system. In addition, the negligible loss of GA and DMB in the dark controls suggests there was negligible evaporation of the precursor or the photosensitizer during the experiments.

### 2.2 Chemical and Optical Analyses

The concentrations of GA and 3,4-DMB were determined by a high-performance liquid chromatograph equipped with a

diode array detector (HPLC-DAD, Agilent Technologies Inc.). A ZORBAX Eclipse XDB-$C_{18}$ column (150×4.6 mm, 5 μm) was used with a mobile phase consisting of a mixture of acetonitrile and water (20:80), and the flow rate was 0.7 mL min$^{-1}$.

Both GA and DMB were detected at 280 nm, and their retention times were 8.349 and 15.396 min, respectively. The mass concentration and chemical composition of the aqSOA products were characterized using a high resolution time-of-flight aerosol mass spectrometer (HR-AMS; Aerodyne Res. Inc). The liquid samples were atomized in argon (Ar, industrial grade, 99.997 %) followed by diffusion drying (Jiang et al., 2021). This process allowed volatile and semi-volatile products to evaporate, leaving only the low-volatility products in the particle phase, which were characterized by AMS.

HR-AMS data were processed using standard toolkits (SQUIRREL v1.56D and PIKA 1.15D). Since Ar was used as the carrier gas, the $CO^+$ signal of aqSOA was quantified directly (Yu et al., 2016). While the organic $H_2O^+$ signal (org-$H_2O^+$) can also be directly determined for dry aerosols, it tends to be noisy due to high sulfate $H_2O^+$ ($SO_4$-$H_2O^+$) signal interference. Therefore, org-$H_2O^+$ was parameterized as org-$H_2O^+$ = $0.4 \times CO_2^+$, based on the linear regression between the determined org-$H_2O^+$ signal (= measured-$H_2O^+$ − $SO_4$-$H_2O^+$) and the measured organic $CO_2^+$ signal (Jiang et al., 2021). The other org-$H_2O^+$ related signals were parameterized as org-$OH^+$ = $0.25 \times$org-$H_2O^+$ and org-$O^+$ = $0.04 \times$org-$H_2O^+$ (Aiken et al., 2008). Atomic ratios of oxygen-to-carbon (O/C) and hydrogen-to-carbon (H/C), and organic mass-to-carbon ratio (OM/OC) ratios, were subsequently determined (Aiken et al., 2008), and the average oxidation state of carbon ($OS_C$) of aqSOA was calculated as $OS_C$ = $2 \times$O/C-H/C (Kroll et al., 2011). The aqSOA concentration in the solution ([Org]$_{solution}$, µg mL$^{-1}$) was calculated using sulfate as the internal standard:

$$[\text{Org}]_{\text{solution}} = [\text{Org}]_{\text{AMS}} \times \frac{[\text{Sulfate}]_{\text{solution}}}{[\text{Sulfate}]_{\text{AMS}}} \qquad \text{(Eq. 1)}$$

where [Org]$_{AMS}$ and [Sulfate]$_{AMS}$ are the AMS-measured concentrations (µg m$^{-3}$) of aqSOA and sulfate in the aerosolized solution, and [Sulfate]$_{solution}$ is the spiked concentration (µg mL$^{-1}$) of sulfate in the solution. The aqSOA mass yield ($Y_{SOA}$) after a given time of illumination (t) was calculated as:

$$Y_{\text{SOA}} = \frac{[\text{Org}]_t}{[\text{GA}]_0 - [\text{GA}]_t} \qquad \text{(Eq. 2)}$$

where [GA]$_0$ is the initial GA concentration (µg mL$^{-1}$) in the solution, and [Org]$_t$ and [GA]$_t$ denote the concentrations of the aqSOA and GA, respectively, in the solution after a period of irradiation.

The light absorbance of the reaction solution was measured using a UV-Vis spectrophotometer (UV-2501PC, Shimadzu). The mass absorption coefficient, the absorption Ångström exponent, and the rate of sunlight absorption of the aqSOA were calculated (Section S1).

Positive matrix factorization (PMF) was performed on the combined matrix of the high-resolution mass spectra (*m/z* 12–360) and the UV-vis spectra (280–600 nm) of the •OH-aqSOA and $^3$C*-aqSOA separately (Jiang et al., 2023). The PMF results were evaluated using the PMF Evaluation Toolkit (PET v3.08 downloaded from: http://cires1.colorado.edu/jimenez-group/wiki/index.php/PMF-AMS_Analysis_Guide). A three-factor solution with fPeak = 0 was chosen based on the evaluation criteria (Ulbrich et al., 2009; Zhang et al., 2011) for •OH- and $^3$C*-aqSOA. Figures S1 and S2 summarize the diagnostic plots for the 3-factor PMF solutions for •OH- and $^3$C*-aqSOA, respectively.

## 3. Results and Discussion

### 3.1 Formation and Characteristics of the aqSOA from Photooxidation of GA by •OH and $^3C^*$

Figures 1a and 1h demonstrate that the loss of GA follows first-order kinetics in both •OH- and $^3C^*$-mediated photoreactions. The pseudo-first-order rate constants were determined to be 0.14 and 0.73 $h^{-1}$, respectively, under our experimental conditions. Based on a previous study (Smith et al., 2016), direct photolysis of GA is expected to be negligible in this study. The fact that the reaction of GA with $^3C^*$ is much faster than with •OH is consistent with previously reported kinetics for other phenols (Smith et al., 2014; Yu et al., 2016) and can be attributed to the higher oxidant concentration in the

$^3C^*$-mediated reaction. Based on the second-order rate constants for GA reacting with •OH ($1.5\times10^{10}$ $M^{-1}$ $s^{-1}$) (Arciva et al., 2022) and with $^3C^*$ ($1.8\times10^9$ $M^{-1}$ $s^{-1}$) (Ma et al., 2021), we estimate that the steady-state concentrations of oxidants are: [•OH] = $2.6\times10^{-15}$ M under the •OH-mediated reaction condition and [$^3C^*$] = $1.1\times10^{-13}$ M under the $^3C^*$-mediated reaction condition. These oxidant concentrations are comparable to previously observed values in fog waters ([•OH] ~ $2.0\times10^{-15}$ M and [$^3C^*$] ~ $5.0\times10^{-14}$ M) (Kaur and Anastasio, 2017, 2018) and in water extracts of ambient particles ([•OH] ~ $4.4\times10^{-16}$ M and [$^3C^*$] ~

$1.0\times10^{-13}$ M) (Kaur et al., 2019).

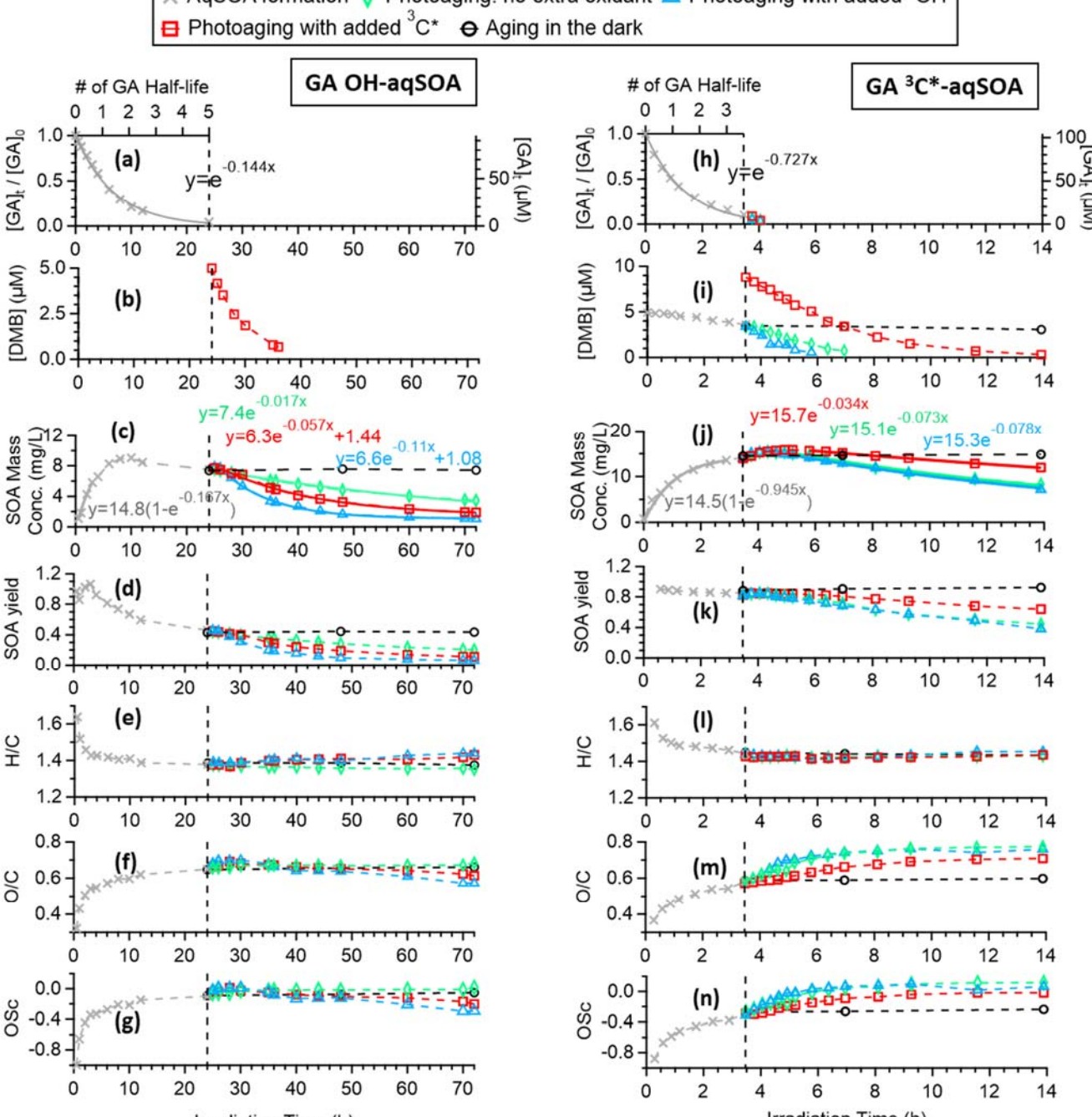

**Figure 1. Overview of aqSOA formation and aging in •OH- and 3C\*-initiated photoreactions of GA. Decay of (a & h) GA and (b & i) 3,4-DMB in the solution. Trends of aqSOA (c & j) mass concentration and (d & k) mass yield and (e & l) H/C, (f & m) O/C and (g & n) OS$_C$ determined by HR-ToF-AMS. These measured values are also shown in Tables S1 and S2.**

As GA is transformed, the mass of the aqSOA increases (Figures 1c and 1j). For both •OH- and $^3$C*-mediated reactions, the aqSOA formation rate relative to the GA decay rate is similar initially, giving a relatively constant mass yield of ~ 90% until one GA half-life ($t_{1/2}$, which is 4.8 h for the •OH reaction and 0.95 h for the $^3$C* reaction). However, in the •OH reaction, the formation of aqSOA slows down after $t_{1/2}$, resulting in a reduction in SOA yield to as low as 46% when ~ 95% of the initial GA has reacted (Fig. 1d). In contrast, in the $^3$C* reaction, the aqSOA yield stabilizes in the range of 85–90% until GA has been

completely consumed (Figures 1d and 1k). These results suggest that the aqSOA reacts with •OH to produce volatile products, which leads to mass loss and slower mass growth. In addition, the results suggest that the photodegradation of •OH-aqSOA of GA has a higher tendency than $^3$C*-aqSOA to form volatile and semi-volatile compounds that evaporate from the condensed phase. This finding is confirmed by prolonged photoaging experiments, which are presented in Sections 3.2 and 3.4.

        The chemical composition of GA aqSOA changes continuously during photoreaction. In both the •OH and $^3$C* reactions,

the O/C, OM/OC, and $OS_C$ of the aqSOA increase, while H/C slightly decreases until all the GA has been consumed (Figures 1e-g and l-n). The HR-AMS spectra of the aqSOA (Figures 2a and 2b) show that when ~95% of the initial GA has reacted (at 24 h for the •OH reaction and 3.5 h for the $^3$C* reaction), the •OH-aqSOA (O/C = 0.64 and $OS_C$ = −0.10) is more oxidized than the $^3$C*-aqSOA (O/C = 0.56 and $OS_C$ = −0.29). In addition, compared with the •OH-aqSOA, the $^3$C*-aqSOA spectrum shows a significantly greater abundance of high m/z ions (Figures 2c-e), including the marker ions of GA oligomers (e.g., $C_{18}H_{19}O_5^+$

and $C_{20}H_{22}O_6^+$ at m/z 315 and 358, respectively) (Jiang et al., 2021), suggesting a higher production of oligomers with $^3$C*. This observation aligns with the trend we observed previously in the aqueous-phase oxidations of phenol and methoxyphenols, where more oligomerization occurred in photoreactions initiated by $^3$C*, while •OH reactions promoted the breakdown of aromatic rings and formation of smaller organic acids (Sun et al., 2010; Yu et al., 2014).

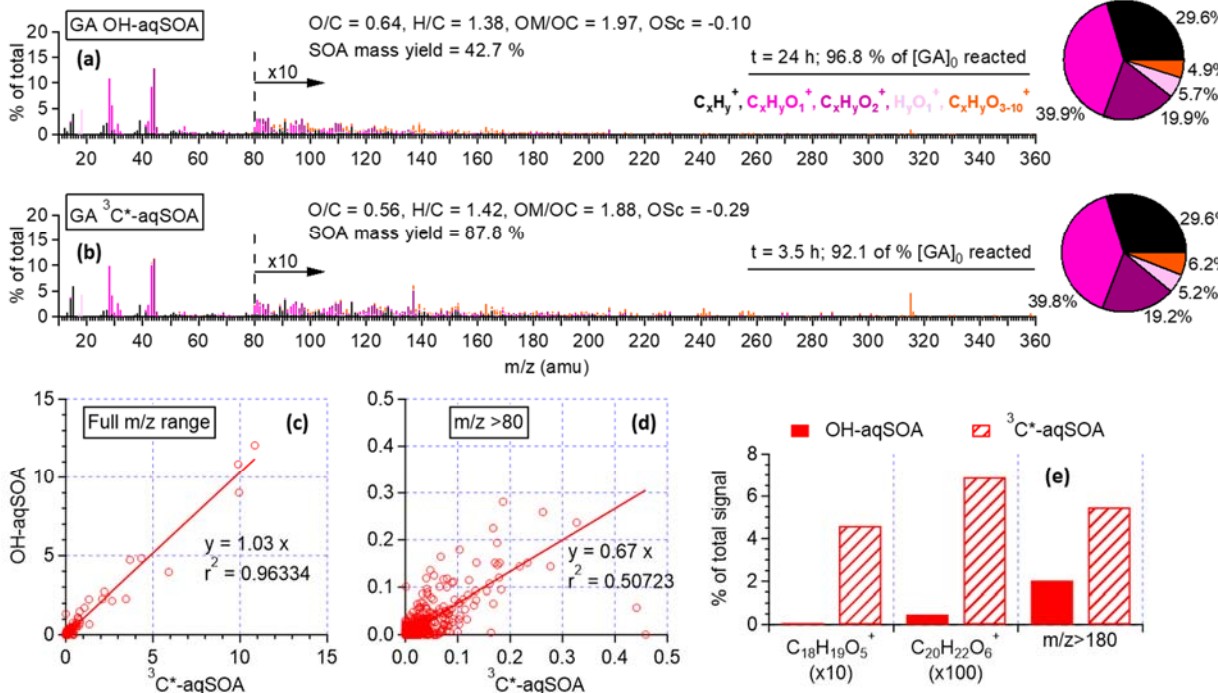

**Figure 2.** HR-AMS mass spectra of (a) •OH-aqSOA and (b) $^3$C*-aqSOA after nearly all the initial GA has reacted. Scatter plots that compare the mass spectra of the •OH-aqSOA with $^3$C*-aqSOA for (c) all ions and for (d) ions with m/z > 80. (e) Relative abundances of the GA-oligomer tracer ions and high mass ions (m/z>180) in the HR-AMS spectra of the aqSOA.

The compositional differences between the •OH-aqSOA and $^3$C*-aqSOA can be attributed to their different reaction mechanisms (as discussed in section 3.2). In the •OH experiment, the reaction can start either by •OH-addition to the aromatic ring to generate OH-adducts or by H-atom abstraction from the hydroxyl group to generate a phenoxy radical. The subsequent coupling of phenoxy radicals leads to the formation of oligomers (Kobayashi and Higashimura, 2003). According to previous studies on phenol oxidation in the gas phase (Atkinson, 1986; Olariu et al., 2002), it has been observed that at room temperature, only ~10% of the phenol + •OH reaction involves H-atom abstraction that leads to the formation of phenoxy radical, whereas ~90% of the •OH reaction proceeds through OH addition. Moreover, modeling studies have indicated that in both gas-phase and aqueous-phase •OH oxidation of phenols, the OH addition pathways exhibit considerably lower activation energy than the H-abstraction pathway (Kılıç et al., 2007). As a result, it is highly likely that the primary products of the •OH reaction with phenols are hydroxyphenols. On the other hand, the $^3$C* reaction primarily proceeds through electron transfer and/or H-atom abstraction which produces a phenoxy radical (Anastasio et al., 1997; Canonica et al., 2000; Yu et al., 2014). The more pronounced production of phenoxy radicals in the $^3$C* reaction can lead to more prominent oligomerization. Although the reactions of phenols with $^3$C* also produce •OH radical, the amount generated is relatively small (Anastasio et al., 1997; Smith et al., 2014) and the •OH addition pathway in the $^3$C* reaction is expected to be less important than in the •OH reaction.

Figures 3d and 3j (and Figures S3a and S4a) show the mass absorption coefficient spectra of the •OH-aqSOA and the $^3$C*-aqSOA. Both aqSOAs are more light-absorbing than the parent GA (Figure S5), which is likely due to the formation of GA oligomers and functionalized products containing conjugated structures. Phenolic dimers and higher oligomers formed through the coupling of phenoxyl radicals and monomeric phenol derivatives formed through •OH and carbonyl addition to the aromatic ring are effective light absorbers (Jiang et al., 2021; Misovich et al., 2021; Yu et al., 2014). In addition, the $^3$C*-aqSOA exhibits greater light absorption than the •OH-aqSOA for a similar extent of GA decay, reflecting the fact that the $^3$C*-aqSOA is generally enriched with more high-molecular-weight conjugated species.

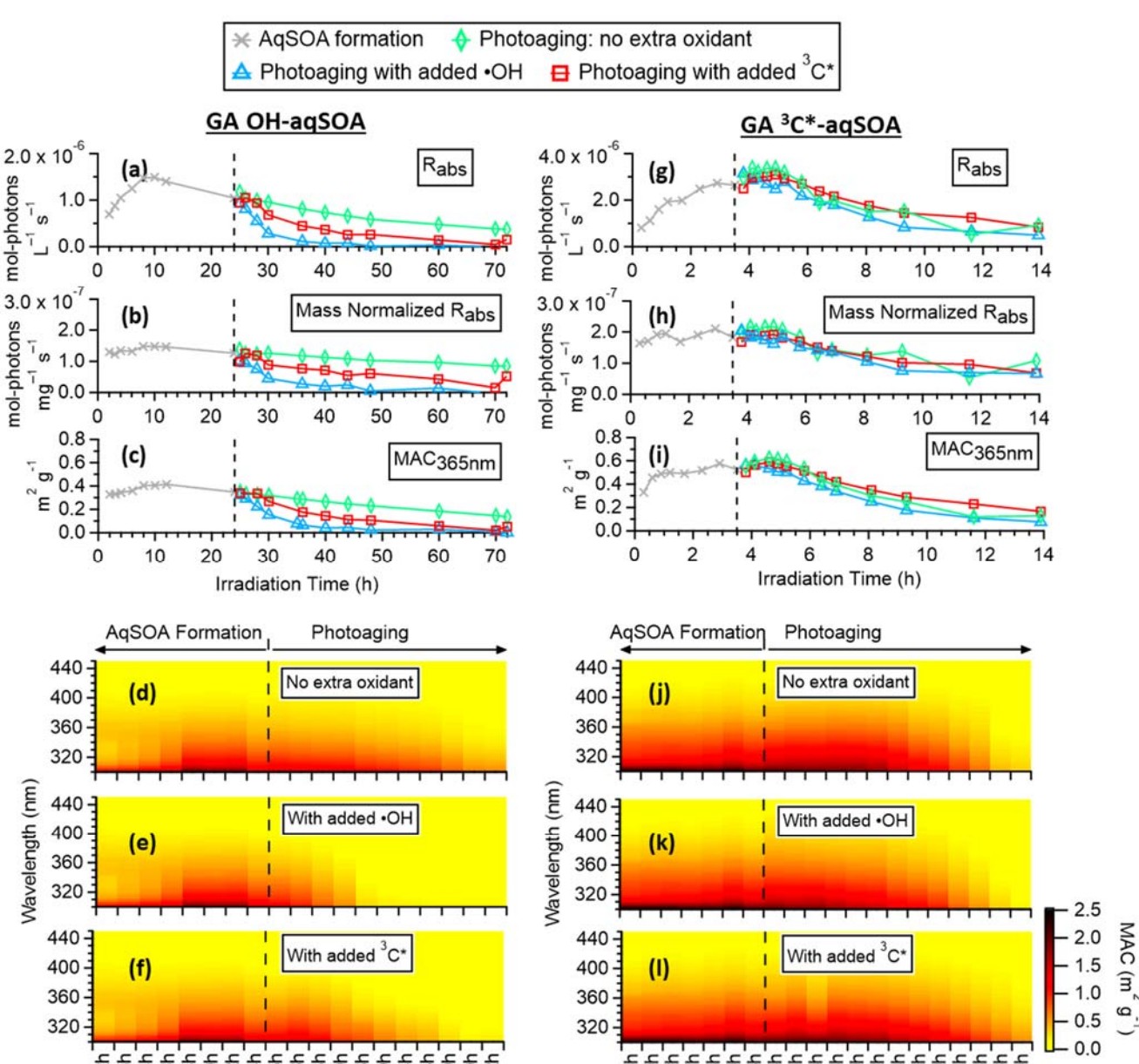

**Figure 3. Evolution of the optical properties of •OH-aqSOA and ³C\*-aqSOA during the course of the photoreactions: (a & g) rate of sunlight absorption; (b & h) rate of sunlight absorption normalized by aqSOA mass; (c & i) mass absorption coefficient (MAC) at 365 nm; and (d-f & j-l) MAC spectra in the wavelength range of 300-450 nm.**

## 3.2 Aqueous-phase Reaction Pathways of Guaiacyl Acetone

Drawing from the results of this study, our previous research (Jiang et al., 2021), and existing literature, we present in Schemes 1 and 2 proposed chemical mechanisms for the aqueous-phase reactions of GA with •OH and ³C\*. As a phenolic

carbonyl, GA has two reactive sites, i.e., the phenol functional group and the carbonyl functional group. Scheme 1 outlines the main reaction pathways triggered by the phenol functional group of GA. In •OH-mediated reactions, •OH can either abstract a H atom from the hydroxyl group to form a phenoxyl radical or add to the aromatic ring to form a dihydroxycyclohexadienyl radical (Olariu et al., 2002). The phenoxyl radical can couple to produce dimers and higher oligomers (Sun et al., 2010), or

react with $HO_2\bullet$ to produce quinonic and hydroxylated products (D'Alessandro et al., 2000). The dihydroxycyclohexadienyl radical reacts with $O_2$ to form a peroxyl radical which can subsequently eliminate a $HO_2\bullet$ to produce hydroxylated products (Barzaghi and Herrmann, 2002). Furthermore, the peroxyl radical can undergo further $O_2$ addition and cyclization to generate a bicyclic peroxyl radical, leading to the cleavage of the aromatic ring via C-C and O-O bond scission and producing fragmented products such as small carboxylic acids, aldehydes, and ketones (Dong et al., 2021; Suh et al., 2003).

In $^3C^*$-mediated reactions, $^3C^*$ can oxidize GA via H-atom abstraction/electron transfer to form a phenoxyl radial and/or a ketyl radical (Anastasio et al., 1997; Smith et al., 2014; Yu et al., 2014). The  ketyl radical can react with $O_2$ to produce superoxide/hydroperoxyl radical ($O_2\bullet^-/HO_2\bullet$), which subsequently react to produce $H_2O_2$ (Anastasio et al., 1997). The photolysis of $H_2O_2$ can serve as a source of •OH in the aqueous phase. Additionally, energy transfer from $^3C^*$ to ground state $O_2$ can lead to the formation of $^1O_2^*$ (Anastasio and McGregor, 2001; McNeill and Canonica, 2016; Zepp et al., 1977).

However, according to our previous studies (Smith et al., 2014; Yu et al., 2014), the amount of •OH and $^1O_2^*$ generated in the reaction of phenols with $^3C^*$ is small and they are expected to be minor oxidants compared to $^3C^*$.

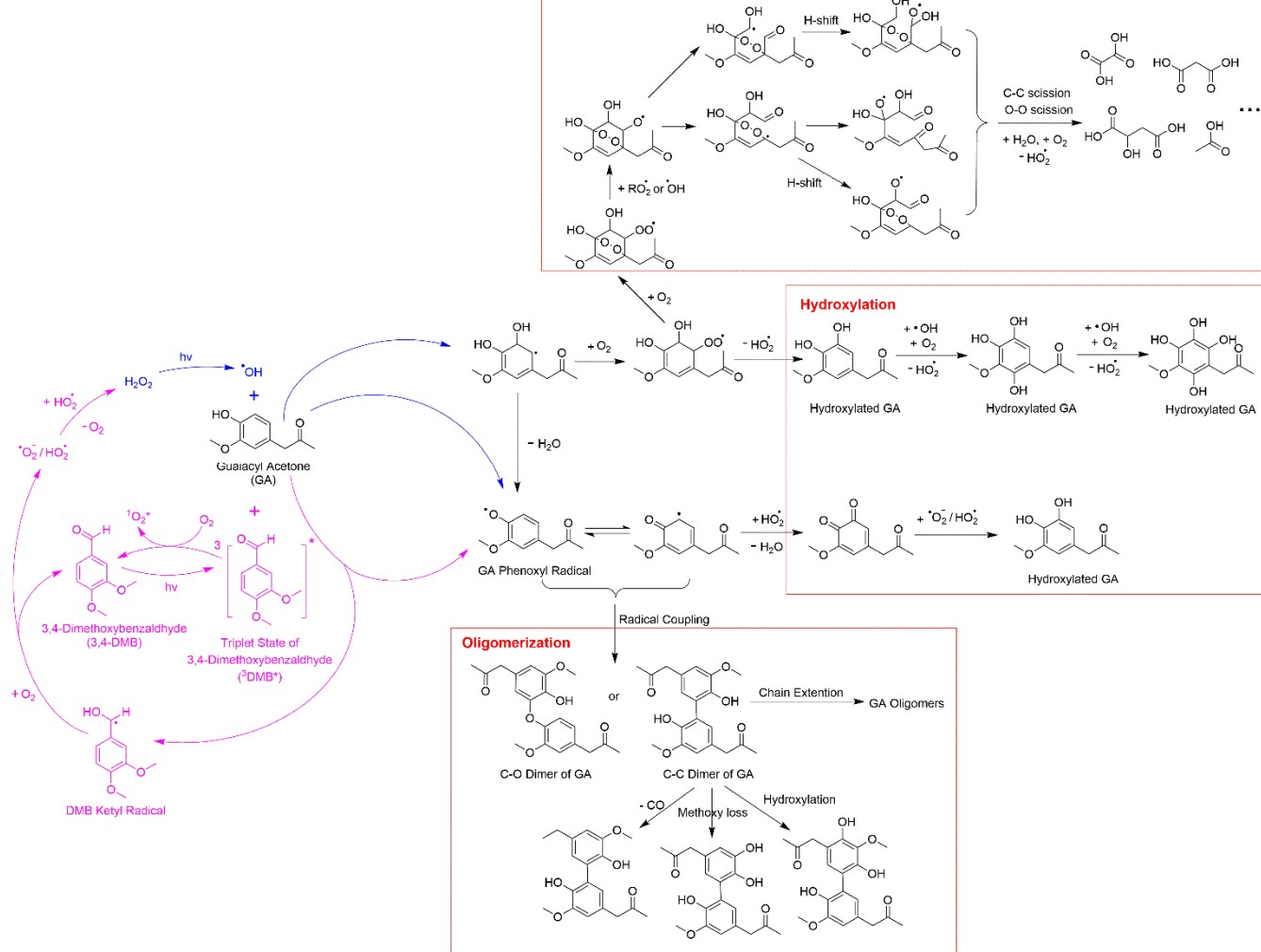

**Scheme 1. Postulated aqueous-phase reaction pathways triggered by the phenol functional group of GA.**

Scheme 2 outlines the reaction pathways that can be triggered by the carbonyl functional group of GA. The α-position of

245 the ketone group of GA can undergo H-atom abstraction by •OH or $^3C^*$, which generates alkyl radicals (Talukdar et al., 2003; Wagner and Park, 2017). The alkyl radicals can react with $O_2$ to produce peroxyl radicals which can further react to form dicarbonyls (Kamath et al., 2018). These dicarbonyls can then undergo photo-dissociation or hydration in the aqueous phase, forming diols and tetrols, which can further react to produce oligomers and functionalized products (Lim et al., 2013; Parandaman et al., 2018; Tan et al., 2010; Zhang et al., 2022). In our previous study (Jiang et al., 2021), we observed that the

250 majority of the GA aqSOA products were formed through reactions triggered by the phenol functional group. The importance of the reactions initiated by the carbonyl functional group may need to be evaluated in future work.

**Scheme 2. Postulated aqueous-phase reaction pathways triggered by the ketone functional group of GA.**

As shown in the proposed reaction pathways, dissolved $O_2$ plays an important role in the aqueous-phase reactions of GA and can influence the reactions in several ways. Firstly, the presence of $O_2$ is essential for the formation of peroxyl radical, which serves as a crucial intermediate in hydroxylation and ring-opening pathways. Therefore, high $O_2$ concentration in the aqueous phase can lead to enhanced hydroxylation and fragmentation, while suppressing oligomer formation from phenoxyl radical (Dong et al., 2021). Additionally, in $^3C^*$-mediated reactions, the involvement of $O_2$ can generate secondary ROS (e.g., $^1O_2^*$, $O_2^{\bullet-}/HO_2\bullet$, and $\bullet OH$) via energy transfer from $^3C^*$ to ground-state $O_2$ (Zepp et al., 1977), electron transfer from $DMB\bullet^{+/\bullet-}$ to $O_2$ (Dalrymple et al., 2010), and reactions between DMB ketyl radical and $O_2$ (Anastasio et al., 1997). These secondary ROS can act as potential oxidants for GA. For instance, $^1O2^*$ reacts with phenols mainly through 1,4-cycloaddition route to produce quinoic products (Al-Nu'airat et al., 2019; García, 1994), whereas $\bullet OH$ and $O_2\bullet^-/HO_2\bullet$ are important contributors to the hydroxylation and ring-cleavage of phenols. Therefore, the presence of $O_2$ is expected to facilitate functionalization and ring-opening pathways while inhibiting oligomerization in $^3C^*$-initiated reactions.

### 3.3 Photo-transformation of AqSOA and Influence of Prolonged Photoaging on SOA Yield and Composition

After ~95% of the initial GA has reacted, the aqSOA was subjected to additional aging under different conditions: 1) aging in the dark; 2) continued illumination without the addition of extra oxidant; and 3) continued illumination with the addition of an oxidant ($\cdot$OH or $^3$C*). As shown in Figures 1c-g, 1j-n, and S6, the mass concentration, elemental ratios, and HR-AMS spectra of the aqSOA remain unchanged during dark aging, indicating negligible dark chemical reactions. In contrast, continued exposure to simulated sunlight results in a 46% reduction in the mass of $^3$C*-aqSOA over about 10.5 hours of prolonged aging (i.e., 14 hours of irradiation in total). More than 60% of the $\cdot$OH-aqSOA mass is degraded after 48 hours of extended photoaging (i.e., 72 hours of irradiation in total). These observations indicate that phenolic aqSOA is susceptible to photodegradation and that fragmentation reactions and evaporation of volatile products likely play important roles in the photoaging process.

The fitted pseudo-first-order decay rate constant ($k$) is 0.073 h$^{-1}$ for the $^3$C*-aqSOA and 0.017 h$^{-1}$ for the $\cdot$OH-aqSOA (Figures 1c, 1j, and 4). The faster decay of the $^3$C*-aqSOA is likely due to the higher oxidant concentration in the $^3$C* reaction during aqSOA aging. Here, we assume that the steady-state concentrations of $\cdot$OH and $^3$C* at the onset of the prolonged photoaging are approximately the same as in the initial solutions, and thus the [$^3$C*] in the $^3$C* reaction is about 40 times higher than the [$\cdot$OH] in the $\cdot$OH reaction during aqSOA aging. This assumption is proved by the first-order decay behavior of GA and the relatively stable 3,4-DMB concentration during the aqSOA formation period. Additionally, $\cdot$OH production from $^3$C* becomes increasingly important during the prolonged photoaging (Anastasio et al., 1997), which may lead to an increased oxidant concentration in the $^3$C* solution. Another possible reason for the faster decay of the $^3$C*-aqSOA compared to the $\cdot$OH-aqSOA may be related to its higher light absorptivity, which can contribute to faster direct photodegradation. However, it is important to note that the rate of photodegradation is also dependent on the quantum yield of photodegradation (i.e., the ratio of the number of compounds destroyed to the number of photons absorbed) (Smith et al., 2016).

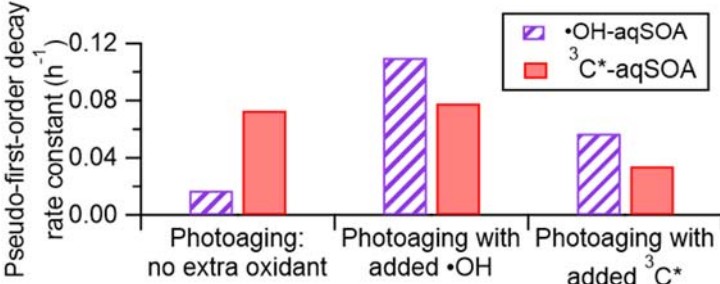

**Figure 4. Pseudo-first-order decay rate constants for loss of mass of $\cdot$OH-aqSOA and $^3$C*-aqSOA under different photoaging conditions**

As depicted in Figures 1l-n and S7, the chemical composition of $^3$C*-aqSOA evolves continuously during photoaging, with the O/C ratio increasing from 0.59 to 0.77, consistent with previous research demonstrating that SOA becomes more oxidized during chemical aging (Kroll et al., 2015; Yu et al., 2016). In contrast, the O/C and H/C ratios of $\cdot$OH-aqSOA exhibit

negligible changes (O/C = 0.67±0.008 and H/C = 1.36±0.008) during prolonged photoaging (Figures 1e-g and S7), even though the mass of aqSOA decreases significantly. This can be explained by the simultaneous evaporation of highly oxidized volatile compounds and the transformation of less oxidized species into more oxidized, low-volatility products, thereby maintaining relatively constant bulk elemental ratios. Additionally, as shown in Figures 5, S8, and S9, both •OH-aqSOA and $^3$C*-aqSOA show increasing $f_{CHO2+}$ (mass fraction of $CHO_2^+$ in the total organic signal, a tracer of carboxylic acids) and decreasing $f_{C2H3O+}$ (mass fraction of $C_2H_3O^+$, a tracer of non-acid carbonyls) during prolonged photoaging, indicating the importance of acid formation in the aqSOA. Furthermore, the continuous increase of $f_{CHO2+}$ indicates a more pronounced production of carboxylic acids from the $^3$C*-aqSOA compared to the •OH-aqSOA. However, acid formation is comparably important during •OH-aqSOA and $^3$C*-aqSOA formation initially.

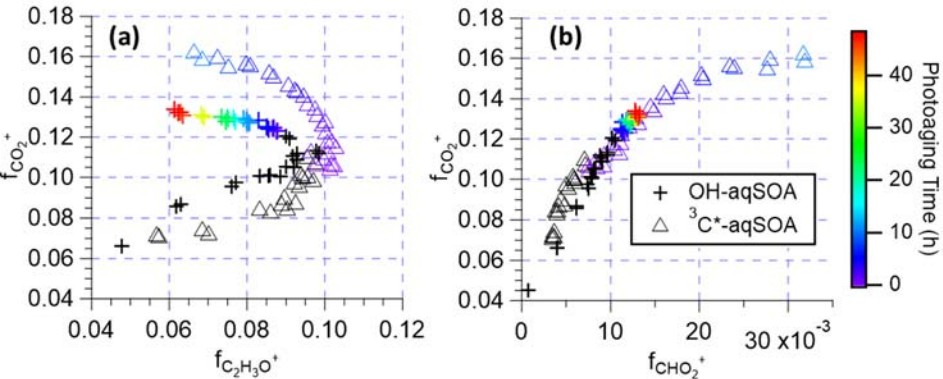

**Figure 5. The plots of $f_{CO2+}$ vs. $f_{C2H3O+}$ and $f_{CO2+}$ vs $f_{CHO2+}$ that illustrate the evolution of the •OH-aqSOA and $^3$C*-aqSOA during the formation and the prolonged photoaging periods under the condition of without extra oxidant addition. The black markers represent the period of aqSOA formation, while the colored markers represent prolonged aqSOA aging (i.e., after ~95% of the initial GA is consumed).**

To further elucidate the chemical evolution of the aqSOA, we performed PMF analysis on the combined AMS and UV–vis absorption spectral data and successfully resolved three distinct factors for both •OH-aqSOA and $^3$C*-aqSOA, each with different temporal profiles, mass spectra, and absorption spectra that represent different generations of aqSOA products (Figures 6 and 7). The formation and decay rate constants of different generations of the aqSOA products were determined by performing exponential fits (y = a(1−e$^{−bx}$) + c and y = ae$^{−bx}$ + c, respectively) to the time trends of the aqSOA factors (Figures 6d,f,h and 7d,f,h). The fitted parameter b (in the unit of h$^{-1}$) represents the first-order rate constant for the aqSOA formation or decay in the photoreactor.

The first-generation $^3$C*-aqSOA, which is the least-oxidized (O/C=0.49 and H/C=1.48), shows enhanced ion signals corresponding to GA oligomers, such as $C_{18}H_{19}O_5^+$ and $C_{20}H_{22}O_6^+$ (Figures 7a, 7j, 7k and S11e-f). These products grow rapidly and peak within the first hour of $^3$C*-aqSOA formation, but they subsequently decrease and disappear completely when GA is consumed. The second-generation factor (O/C=0.59 and H/C=1.42), in which the oligomer tracer ions are substantially reduced, shows enhanced ion signals corresponding to functionalized GA monomers or ring-opening dimers such as $C_9H_7O_3^+$

and $C_{15}H_{11}O_4^+$ (Figures 7b, 7j, 7k and S11d). The second-generation products build up gradually, peak after GA is consumed, and degrade more slowly than the 1st-generation $^3C^*$-aqSOA (k = 0.36 h$^{-1}$ vs 1.8 h$^{-1}$) during prolonged aging (Figure 7d).

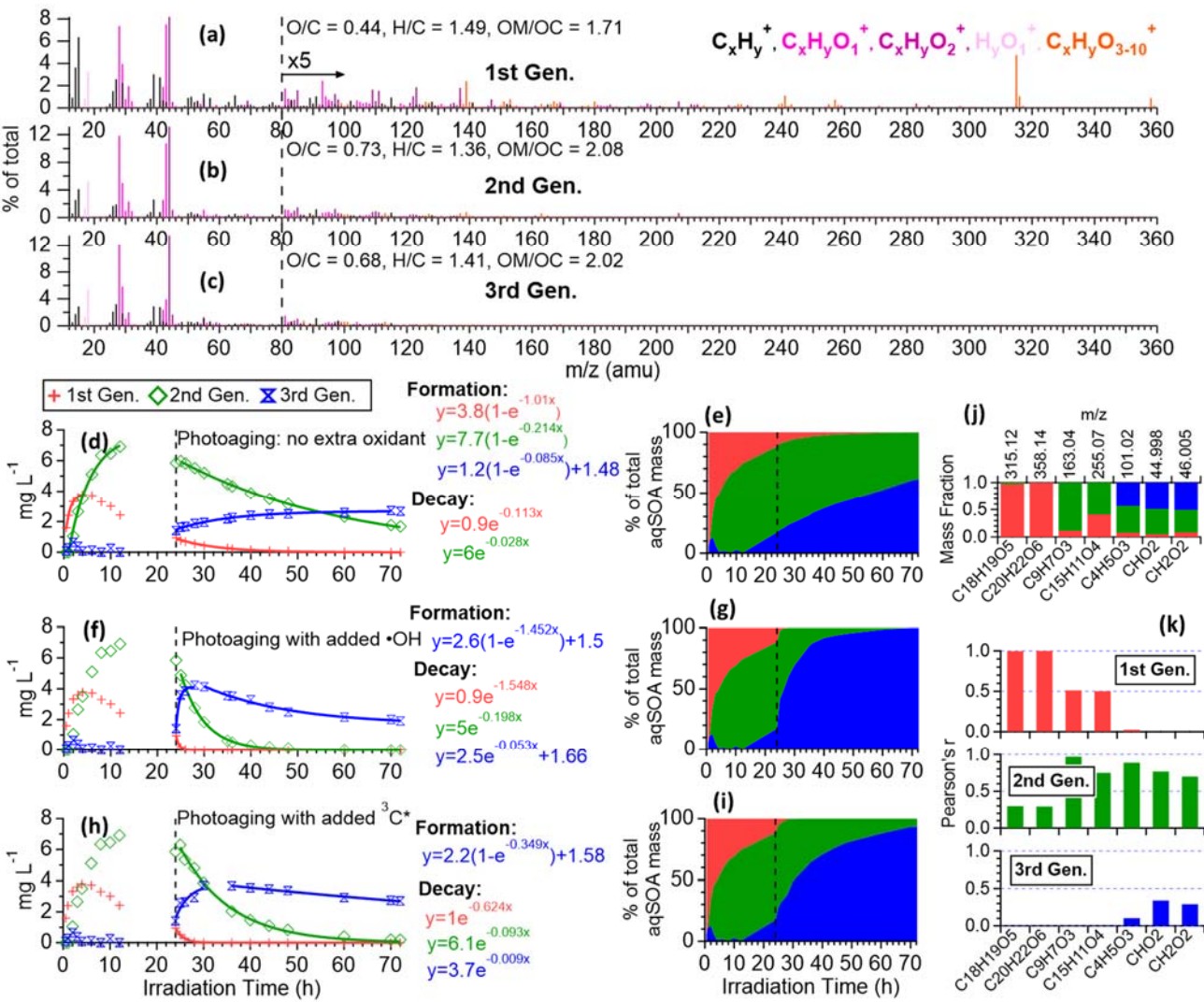

**Figure 6. Characteristics of the three generations of the •OH-aqSOA products resolved by PMF: (a-c) MS profiles; (d, f, and h) mass concentration time series; and (e, g, and i) fractional contribution time series of the PMF factors. (j) Mass fraction of selected AMS tracer ions attributed to each PMF factor. (k) Correlation between PMF factors and selected AMS tracer ions.**

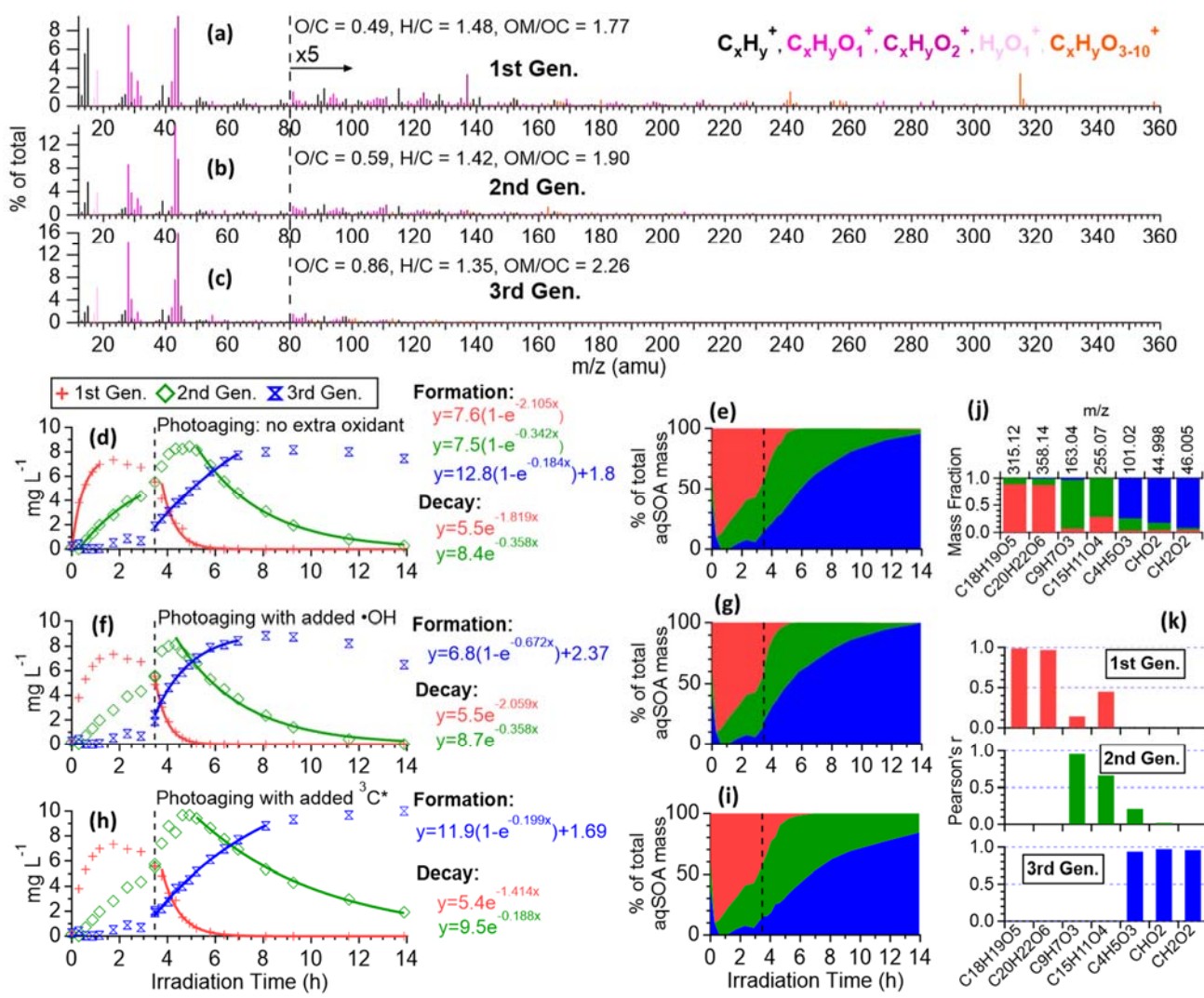

**Figure 7. Characteristics of the three generations of the $^3C*$-aqSOA products resolved by PMF: (a-c) MS profiles; (d, f, and h) mass concentration time series; and (e, g, and i) fractional contribution time series of the PMF factors. (j) Mass fraction of selected AMS tracer ions attributed to each PMF factor. (k) Correlation between PMF factors and selected AMS tracer ions.**

The third-generation factor of $^3C*$-aqSOA is the most oxidized (O/C = 0.86 and H/C = 1.36), and its mass spectrum shows negligible high m/z signals but elevated small, oxygenated ions such as $CHO_2^+$, $CH_2O_2^+$, and $C_4H_5O_3^+$ (Figures 7c, 7j, 7k and S11a-c). The increase of this factor is observed when the 1st-generation factor starts to decline. It continues to increase but shows a slight decrease towards the end of prolonged photoaging. These findings agree with our previous studies, which demonstrate that oligomerization and functionalization play a more significant role in the initial formation of phenolic aqSOA, while fragmentation and ring-opening reactions to produce more oxidized compounds become more important later (Jiang et al., 2021; Yu et al., 2016). Further, the observed decay of the 3rd-generation aqSOA indicates that prolonged aging leads to

the formation of volatile compounds that evaporate from the condensed phase, resulting in mass loss of aqSOA. This implies that photochemical aging can remove aqSOA from the atmosphere, in addition to wet and dry deposition (Hodzic et al., 2016).

The mass spectral features of the •OH-aqSOA factors (Figures 6a-c) are generally similar to those of the $^3$C*-aqSOA factors (Figures 7a-c). However, in •OH-aqSOA, we observed that the 2nd-generation (O/C = 0.73 and H/C = 1.36) is the most oxidized factor and shows strong correlations not only with the tracer ions representing functionalized GA monomers (e.g., $C_9H_7O_3^+$ and $C_{15}H_{11}O_4^+$) but also with a group of small, oxygenated ions (e.g., $CHO_2^+$, $CH_2O_2^+$, and $C_4H_5O_3^+$) that are enriched in the 3rd-generation $^3$C*-aqSOA. One possible reason for the observed difference is that •OH reaction tends to form highly oxidized products that degrade over long aging times, whereas the $^3$C* reaction can generate highly oxidized SOA products that are more resistant to degradation. Another possible explanation for the observed difference in the evolution of $^3$C*-aqSOA and •OH-aqSOA is that the highly oxidized species in the aqSOA exhibit different reactivity with $^3$C* and •OH due to their electron availability (Walling and Gibian, 1965). In general, $^3$C* is known to be less reactive with electron-poor compounds, whereas •OH can rapidly react with a wide range of organic compounds in the aerosol at diffusion-controlled rates (Herrmann et al., 2010). As a result, electron-poor products may persist in $^3$C*-aqSOA, while •OH has the capability to further oxidize these products, eventually transforming them into volatile species. This explanation is supported by the more significant decay of the 3rd-generation $^3$C*-aqSOA when extra $H_2O_2$ is added during prolonged photoaging (Figures 7d and 7f). In addition, compared to $^3$C*-aqSOA, •OH-aqSOA exhibits much lower production of 1st-generation products but higher 2nd-generation (Figures 7d-e vs. Figures 6d-e), suggesting that oligomerization is more pronounced in $^3$C*-aqSOA, while functionalization plays a more important role in •OH-aqSOA.

### 3.4 Evolution of AqSOA Optical Properties during Prolonged Aging

Figures 3, S3, and S4 illustrate the evolution of the light absorption properties of the aqSOA during formation and aging. The aqSOA experiences photobleaching during prolonged aging, with the $MAC_{365nm}$ value of the •OH-aqSOA decreasing from 0.41 (the maximum) to 0.14 $m^2\,g^{-1}$, and that of $^3$C*-aqSOA decreasing from 0.62 (the maximum) to 0.13 $m^2\,g^{-1}$. The rates of sunlight absorption, both normalized and un-normalized by aqSOA mass, also decrease during prolonged aging. Figure 8 displays the absorption Ångström exponent (AAE) of the aqSOA as a function of $\log_{10}(MAC_{405})$ and an optical-based classification of BrC (Saleh, 2020; Zhai et al., 2022). As a result of prolonged photoaging, the GA aqSOA shifts from being classified as weak BrC to very weak BrC. The changes in the light absorption properties of the GA aqSOA are also influenced by elevated oxidant concentrations (see Section 3.4 for further discussions).

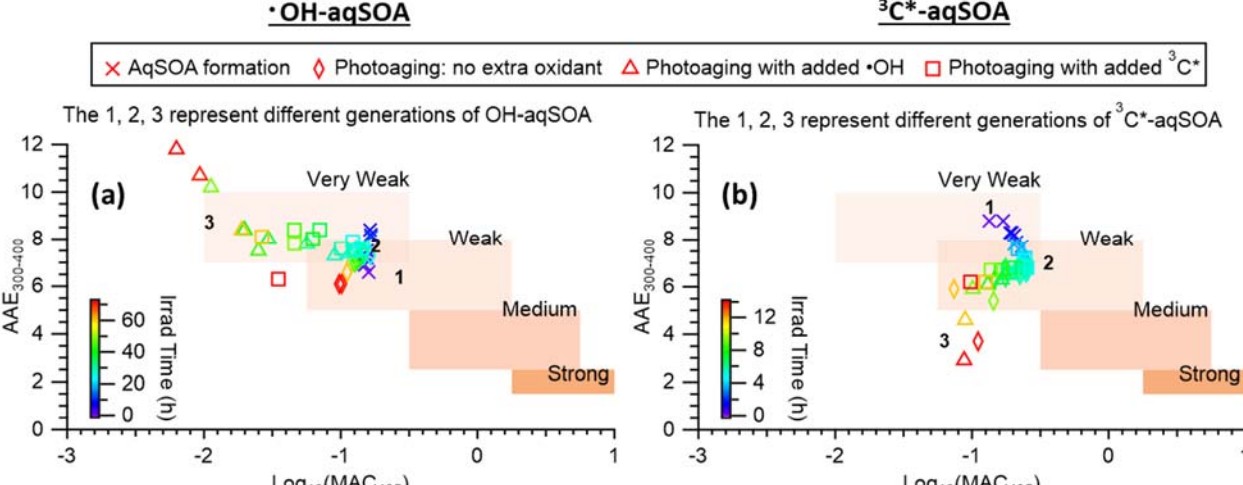

**Figure 8. The light absorption properties of (a) •OH-aqSOA and (b) ³C\*-aqSOA as shown in the AAE vs. log₁₀(MAC₄₀₅) space. The shaded areas in each plot represent very weakly, weakly, moderately, and strongly absorbing BrC denoted based on the optical-based BrC classification scheme (Saleh, 2020; Zhai et al., 2022). The numbers 1, 2, and 3 represent the different generations of the •OH-aqSOA and the ³C\*-aqSOA products obtained from PMF.**

Figure 9 presents the mass absorption coefficient spectra resolved by PMF for the three generations of GA aqSOA resulting from •OH and ³C\* reactions. In general, the ³C\*-aqSOA factors are more light-absorbing than the •OH-aqSOA factors, which is consistent with the higher abundance of oligomers and conjugated high molecular weight products in the ³C\*-aqSOA. The 1st-generation aqSOA factor exhibits a hump in the MAC spectra between 340 and 400 nm, a feature observed previously in phenolic aqSOA (Smith et al., 2016) and attributed to the high conjugation present in oligomeric products. For both •OH- and ³C\*-mediated reactions, the intermediate, 2nd-generation aqSOA are the most light-absorbing compared to the fresher (i.e., 1st-generation) and more aged (i.e., 3rd-generation) aqSOA. Nevertheless, the 2nd-generation •OH-aqSOA shows relatively lower MAC values (MAC$_{365nm}$ = 0.47 m$^2$ g$^{-1}$) than the 2nd-generation ³C\*-aqSOA (MAC$_{365nm}$ = 0.89 m$^2$ g$^{-1}$). This difference could be attributed to the more pronounced oxidative ring-opening reactions that cause the destruction of conjugation in •OH-aqSOA, resulting in the breakdown of chromophores. The 3rd-generation aqSOA factors are the least absorbing (MAC$_{365nm}$ = 0.070 m$^2$ g$^{-1}$ for the •OH-aqSOA and 0.018 m$^2$ g$^{-1}$ for the ³C\*-aqSOA), consistent with the dominance of fragmented and ring-opening products in prolonged aging.

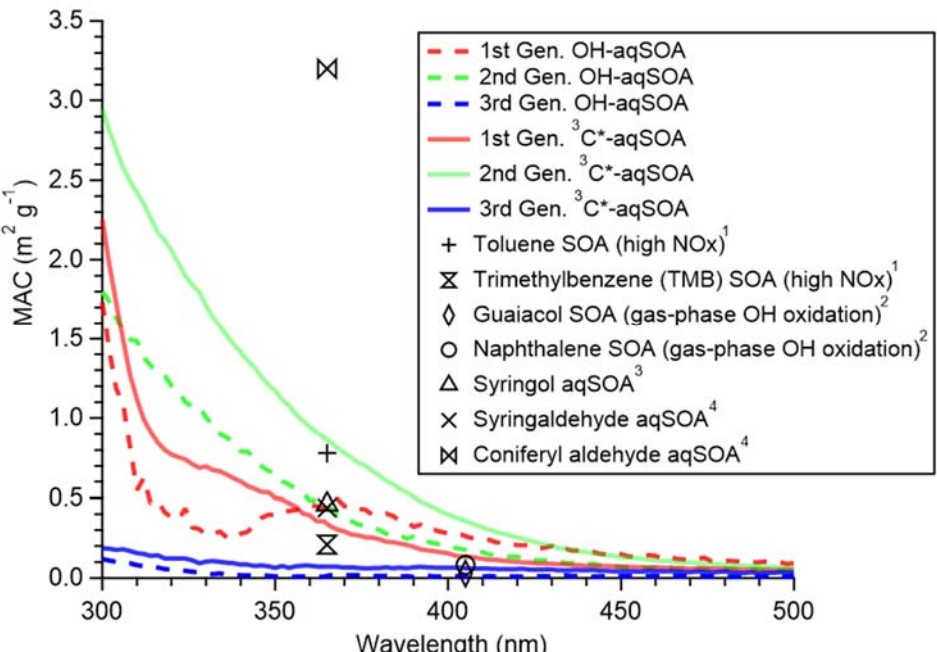

**Figure 9. Mass absorption coefficient spectra of the PMF resolved three generations of the OH-aqSOA and the [3]C*-aqSOA, comparing with previously reported MAC values of SOA produced from aromatic precursors by [1] Liu et al., 2016, [2] Lambe et al., 2013,[3] Yu et al., 2014 and [4] Smith et al., 2016.**

### 3.5 Effects of Additional Oxidant Exposure on AqSOA Aging

To investigate the effect of the concentrations of condensed-phase oxidants on the photoaging of phenolic aqSOA, we added either 100 μM $H_2O_2$ or 5 μM 3,4-DMB into the solution after the majority (~ 95 %) of GA had reacted. Since GA decay follows first-order kinetics, we assumed that the steady-state concentration of oxidants remained constant during initial aqSOA formation. By introducing additional $H_2O_2$ or 3,4-DMB, we increased the •OH or [3]C* concentration, as well as the overall oxidant concentration in the solution during the photoaging of the aqSOA.

As shown in Figures 1c and 4 and Table S3, compared to continued photoaging without addition of extra oxidant ($k$ = 0.017 h[-1]), the photodegradation rates of •OH-aqSOA are substantially faster when extra •OH or [3]C* are introduced ($k$ = 0.11 and 0.057 h[-1], respectively). Likewise, the addition of extra •OH or [3]C* results in more extensive mass loss of the •OH-aqSOA with reductions of 88% or 79% of the aqSOA mass observed at the end of the photoaging, respectively. These levels of mass loss were significantly higher than without extra oxidant (i.e., 62%). These findings suggest that the presence of additional

•OH or [3]C* accelerates the photochemical aging process and leads to increased formation of volatile and semi-volatile products that subsequently evaporate. As shown in Figures 6d, 6f, and 6h, the decay of the 1st- and 2nd-generation •OH-aqSOA is increased, and concurrently, the formation of the 3rd-generation factor is accelerated when extra oxidants are introduced,

suggesting a faster transformation from the 1st to 2nd to 3$^{rd}$ generation. In addition, in the later stage of photoaging, we observed a more significant decay of the 3rd-generation •OH-aqSOA when extra oxidants are added, which suggests that higher concentrations of oxidants also facilitate the ultimate breakdown of the 3rd-generation •OH-aqSOA. The O/C and the average oxidation state of carbon (OSc) of •OH-aqSOA exhibit a slightly faster increase upon the addition of extra •OH or $^3$C*, but eventually decrease more significantly by the end of the photoaging (Figures 1f and 1g), indicating accelerated formation of highly oxidized species and enhanced production of volatile compounds under high oxidant concentrations. Furthermore, increased oxidant concentrations also have significant impacts on the photobleaching of •OH-aqSOA. Specifically, the MAC values of the aqSOA decrease faster when extra oxidants are added (Figure 3c-f). It is noteworthy that the addition of 100 μM of $H_2O_2$, the source of •OH, has a greater impact on the degradation of •OH-aqSOA mass and light absorption than the addition of 5 μM of 3,4-DMB, the source of $^3$C*, despite the fact that the GA reaction with $^3$C* is much faster than that with •OH in this study. This result may reflect the reactivity differences between GA and the aqSOA towards the oxidants, i.e., while both oxidants react quickly with GA, for the electron-poor aqSOA products, $^3$C* may be much less reactive whereas •OH can oxidize them rapidly. Another possible interpretation is that the addition of 3,4-DMB into the solution during aqSOA aging produces unique low-volatility, light-absorbing products which cannot be generated in •OH reactions. These products may counteract some of the mass and absorption loss due to fragmentation and evaporation.

Elevated oxidant concentrations also affect the photoaging of $^3$C*-aqSOA. Unlike •OH-aqSOA, the addition of •OH during aging only slightly accelerates the decrease of mass and light absorption of the $^3$C*-aqSOA (Figure 1j-k and Figure 3g-l). One possible explanation is that the added •OH only accounts for a small fraction of the total oxidant amount in the $^3$C*-initiated reaction system, and thus shows little impact on the aqSOA aging compared to the preexisting oxidants (e.g., $^3$C* and •OH generated from $^3$C*). Another possible explanation is that the added •OH reacts with 3,4-DMB in the solution, resulting in a decrease in the amount of $^3$C* source. Additionally, the reaction of •OH with 3,4-DMB may also generate low-volatility products that balance out the increased decay of the aqSOA. This interpretation is consistent with the fast decay of 3,4-DMB after the addition of extra •OH (Figure 1i). However, when extra $^3$C* (i.e., 3,4-DMB) is added to the $^3$C*-aqSOA solution during extended aging, it slows down the decay of $^3$C*-aqSOA mass and light absorption due to the enhanced formation of 2nd-generation products (Figure 7h vs. Figure 7d). This suggests that an increase in $^3$C* concentration during aging promotes the formation of low-volatility functionalized products.

## 4. Conclusions

This study investigates the evolution of the composition and optical properties of phenolic aqSOA during prolonged photoaging, including the effects of increased oxidant concentrations. The aqSOA was generated by reacting GA with •OH or $^3$C* under relatively dilute cloud/fog water conditions. Compared to the •OH reaction, the $^3$C* reaction is significantly faster, and leads to higher mass yields of aqSOA, more oligomers, and increased high molecular-weight species when the same fraction of initial GA has reacted. On the other hand, the •OH reaction generates aqSOA that is more oxidized and more

enriched in small, highly oxygenated species. Consistent with their compositional differences, the $^3$C*-aqSOA is more light-absorbing than the •OH-aqSOA.

The chemical composition of the aqSOA evolves during photoaging, with oligomerization and functionalization being the dominant mechanisms during initial aqSOA formation, while fragmentation and volatile product formation becoming more important during prolonged aging. This leads to a loss of 62–88% of the •OH-aqSOA mass after 48 hours of prolonged aging under simulated sunlight with or without added oxidants, while the $^3$C*-aqSOA experienced a loss of 25–54% of its mass after 10.5 hours of extended photoaging. These results indicate that aqueous-phase photochemical aging can significantly reduce atmospheric aqSOA, in addition to wet and dry deposition. In this study, the rate of loss for phenolic aqSOA during photochemical aging was found to be in the range of 0.017–0.11 h$^{-1}$ (i.e., 5–30×10$^{-6}$ s$^{-1}$). The photochemical kinetics in our RPR-200 photoreactor system were ~7 times faster than those experienced under ambient winter solstice sunlight in Northern California (George et al., 2015). Consequently, these observations indicate a photochemical lifetime of 3–17 days for phenolic aqSOA in ambient conditions. The deposition loss rate constant of submicron particles in the atmosphere, assuming wet deposition is the dominant process, is approximately $2×10^{-6}$ s$^{-1}$ (resulting in a lifetime of approximately 5 days) (Henry and Donahue, 2012; Molina et al., 2004). These findings suggest that the contribution of photochemical aging to the removal of phenolic aqSOA can be comparable to that of wet deposition.

The average oxidation state of the $^3$C*-aqSOA increases continuously during photoaging, while that of •OH-aqSOA exhibits a slight decrease towards the end of photoaging when additional oxidants are introduced; this is likely due to the formation and evaporation of highly oxidized volatile products. This finding indicates that photoaging does not necessarily increase the average oxidation state of condensed-phase organics, as the evaporation of highly oxidized products may decrease the average O/C of aqSOA. As photoaging continues, photobleaching becomes more pronounced, causing the aqSOA to shift from weakly absorbing BrC to very weak BrC. We also observed through PMF analysis that the second-generation aqSOA, enriched in functionalized phenolic compounds, is the most light-absorbing. This suggests that intermediately-aged phenolic aqSOA is more light absorbing than other generations, and that the light absorptivity of phenolic aqSOA is the result of a competition between BrC formation and photobleaching. Elevated oxidant concentrations during photoaging promote fragmentation reactions over oligomerization and functionalization reactions and can ultimately promote the breakdown and evaporation of the aqSOA products, resulting in a faster decline in aqSOA mass and light absorption.

**Acknowledgments**

This research was supported by grants from the National Science Foundation (NSF) (Grant #AGS-2220307), the U.S. Department of Energy (DOE) Atmospheric System Research Program (Grant #DESC0022140), and the California Agricultural Experiment Station (Projects CA-D-ETX-2102-H and CA-D-LAW-6403-RR). W.J. also acknowledges funding from the Fumio Matsumura Memorial Fellowship by the Department of Environmental Toxicology at the University of California at Davis.

**Author contributions**

QZ and WJ developed the research goals and designed the experiments. WJ and CN performed the experiments. WJ and QZ analyzed the data and prepared the manuscript with contributions from all co-authors.

**Competing interests**

The contact author has declared that none of the authors has any competing interests.

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
