# Peer review of "Photoaging of Phenolic Secondary Organic Aerosol in the Aqueous Phase: Evolution of Chemical and Optical Properties and Effects of Oxidants"

_EGUsphere, 2023_

## Author Comment (AC1)

We thank the reviewer for their thoughtful and constructive comments, and we have revised the manuscript accordingly. Listed below are our point-to-point responses (in **blue**) to the comments (in **black**) and changes of the manuscript (in **red**).

**Responses to Reviewer 1:**

Comments:

This study examines the evolution of chemical and optical properties of phenolic aqSOA generated via •OH- or $^3$C* oxidation during photoaging, as well as the effects of increased concentration of oxidants. The changes in the chemical composition and light absorption of aqSOA were tracked using HR-AMS and UV-Vis spectroscopy. The findings for the $^3$C*-aqSOA have been reported in a previous study by the same group (Jiang et al., 2021). Although this paper attempted to explain the differences between $^3$C*-aqSOA thoroughly and •OH-aqSOA, several statements appear unclear or not clearly supported by the results. The paper is well organized, but several points need clarification.

1. Could the authors give examples of products likely resistant to fragmentation?

In our previous studies on phenolic aqSOA formation from syringol, we have observed that certain dimeric products, such as $C_{15}H_{16}O_9$, proposed as 3′,5,5′-Trimethoxy[1,1′-biphenyl]-2,2′,3,4,4′,6′-hexol, are more stable than other oligomeric and functionalized products, and can remain in large abundance after illumination equivalent to several days of atmospheric aging (Yu et al., 2016). We hypothesize that similar dimeric products of GA may also be more resistant to fragmentation. However, due to the lack of molecular-level characterization in this study, we are unable to identify which specific products are more resistant to fragmentation in GA aqSOA.

2. Were the 4 aliquots prepared for further aging also continuously stirred?

Yes, the solutions were continuously stirred during further aging. This information has now been added to Sec. 2.1. of the updated manuscript at line 114.

3. Section 3.2: why would •OH production from $^3$C* be more important during prolonged photoaging? is there any evidence for this?

$H_2O_2$ can be produced via the oxidation of phenols with $^3$DMB*, in which a ketyl radical is formed and subsequently reacts with $O_2$ to produce $HO_2$• that undergoes self-reaction to generate $H_2O_2$. Moreover, it has been found that the $H_2O_2$ concentration increases significantly with illumination time in the reactions of phenols with $^3$DMB* (Anastasio et al., 1997). Thus, it is possible to expect higher $H_2O_2$ and •OH concentrations during prolonged photoaging. We have included this reference in the updated manuscript. Furthermore, we have provided Scheme 1 to illustrate •OH production from the reaction of GA with $^3$C*.

[Figure]

Scheme 1. Postulated reaction pathways triggered by the phenol functional group of GA.

4. Section 3.2: Does higher light absorptivity always correlate with faster direct photodegradation? It seems not as, for example, Smith et al. (2016) attributed the essentially equal reactivity of syringaldehyde and acetosyringone against direct photodegradation to the greater light absorption by syringaldehyde and higher quantum efficiency for loss for acetosyringone. Also, how does this reconcile with the statement in section 3.1 about •OH-aqSOA being more vulnerable to photodegradation than $^3$C*-aqSOA?

Thanks for this comment. We agree that both the light absorptivity and the quantum yield of photodegradation dictate the photodegradation rate of a compound (Smith et al., 2016). We have revised the related discussions in Sec 3.2 of the manuscript as follows.

Line 279: Another possible reason for the faster decay of the $^3$C*-aqSOA compared to the •OH-aqSOA may be related to its higher light absorptivity, which can contribute to faster direct photodegradation. However, it is important to note that the rate of photodegradation is also dependent on the quantum yield of photodegradation (i.e., the ratio of the number of compounds destroyed to the number of photons absorbed) (Smith et al., 2016).

In Sec 3.1, we compared the aqSOA formation from the •OH and $^3$C* reactions of GA. When ~95% of GA reacted, the •OH-aqSOA showed a higher degree of oxidation and a more enhanced production of volatile compounds than the $^3$C*-aqSOA, as evidenced by the higher bulk O/C ratio and lower mass yield of the •OH-aqSOA (Figures 1d, 1f, 1k and 1m). This result suggests that oxidative ring-opening pathways are more promoted in the •OH reaction of GA, whereas oligomerization and functionalization are more pronounced in the $^3$C* reaction during the initial aqSOA formation period. However, the aqSOA's vulnerability to degradation depends on the light absorptivity and the quantum yield of photodegradation of the aqSOA. Therefore, the difference in photo-vulnerability of the two aqSOA types cannot be determined solely based on the initial aqSOA formation pathways. We have revised Sec 3.1 accordingly and the text has now been updated as follows.

Line 181: In addition, the results suggest that the photodegradation of •OH-aqSOA of GA has a higher tendency than $^3$C*-aqSOA to form volatile and semi-volatile compounds that evaporate from the condensed phase.

5. Section 3.2: Could the authors explain why acid formation is more pronounced in the aging of $^3$C*-aqSOA? In section 3.1, it was mentioned that •OH-aqSOA has a greater tendency to form volatile and semi-volatile compounds that evaporate from the condensed phase.

Our HR-AMS analysis revealed that the $^3$C*-aqSOA is less oxidized and has a higher oligomer content compared to the •OH-aqSOA. Since the formation of acids is mainly driven by ring-opening reactions of GA and its derivatives (as depicted in Schemes 1 and 2), this compositional difference suggests that $^3$C*-aqSOA contains a greater concentration of acid precursors, which could potentially explain the more pronounced acid formation observed during its aging.

To illustrate the reaction pathways of GA and the possible product formation, we have included Schemes 1 and 2 in the updated manuscript.

Scheme 2. Postulated reaction pathways triggered by the ketone functional group of GA.

6. Section 3.2: What are the differences between the highly oxidized products from $^3C^*$ and •OH oxidation? Those from $^3C^*$ are stated to be resistant to degradation, while those from •OH are mentioned to degrade over long aging times. Also, why are some of the highly oxidized products less reactive with $^3C^*$ than with •OH, and that they can persist in the aqueous phase?

Our PMF results show that small, oxygen-containing ions are predominant in the 3rd-generation $^3C^*$-aqSOA factor but are enriched in both 2nd- and 3rd-generation •OH-aqSOA. The 2nd-generation •OH-aqSOA factor decays faster and earlier than the 3rd-generation factor, suggesting that the •OH-aqSOA may comprise some highly oxidized products that belong to the 2nd-generation and are susceptible to degradation. However, the highly oxidized products in $^3C^*$-aqSOA are primarily classified as 3rd-generation and are expected to be more resistant to degradation.

However, it should be noted that the interpretation of the PMF results presented above is just one possible explanation. In this study, we were not able to provide molecular information on the highly oxidized products generated from $^3C^*$ and •OH reactions, and thus a direct comparison between them is challenging. We will pursuit a more detailed understanding of the composition and degradation pathways of these products in our future studies. To prevent misinterpretation, the text in the manuscript has been revised as follows.

Line 335: One possible reason for the observed difference is that •OH reaction tends to form highly oxidized products that degrade over long aging times, whereas the $^3C*$ reaction can generate highly oxidized SOA products that are more resistant to degradation.

On the other hand, previous studies have shown that the reactivity of $^3C*$ is sensitive to electron availability (Walling and Gibian, 1965), while •OH can react rapidly with a wide range of organic compounds in the condensed phase at diffusion-controlled rates (Herrmann et al., 2010). Therefore, it is possible that electron-poor products may exhibit insignificant reactivity towards $^3C*$, but they can still undergo further oxidation by •OH. This may also contribute to the degradation of highly oxidized products in the •OH-aqSOA over long aging times, whereas the highly oxidized products in $^3C*$-aqSOA are primarily classified as 3rd-generation and are expected to be more resistant to degradation. The text in the manuscript has been updated as follows for clarification.

Line 337: Another possible explanation for the observed difference in the evolution of $^3C*$-aqSOA and •OH-aqSOA is that the highly oxidized species in the aqSOA exhibit different reactivity with $^3C*$ and •OH due to their electron availability (Walling and Gibian, 1965). In general, $^3C*$ is known to be less reactive with electron-poor compounds, whereas •OH can rapidly react with a wide range of organic compounds in the aerosol at diffusion-controlled rates (Herrmann et al., 2010). As a result, electron-poor products may persist in $^3C*$-aqSOA, while •OH has the capability to further oxidize these products, eventually transforming them into volatile species.

7. Section 3.3: Why would the higher oxidation degree of •OH-aqSOA lead to the destruction of chromophores?

In this context, "oxidation" pertains specifically to the oxidative ring-opening reactions or bond breaking which destroy the conjugation in chromophores. To clarify, the text has been updated as follows in the manuscript.

Line 367: Nevertheless, the 2nd-generation •OH-aqSOA shows relatively lower MAC values ($MAC_{365nm}$ = 0.47 $m^2$ $g^{-1}$) than the 2nd-generation $^3C*$-aqSOA ($MAC_{365nm}$ = 0.89 $m^2$ $g^{-1}$). This difference could be attributed to the more pronounced oxidative ring-opening reactions that cause the destruction of conjugation in •OH-aqSOA, resulting in the breakdown of chromophores.

8. Section 3.4: Why would •OH be more reactive with non-phenolic organic compounds? For example, a study on the oxidation of green leaf volatiles (Richards-Henderson et al., 2015) comprising both non-phenolic and phenolic compounds did not show a general trend in the reactivity with 3C* and •OH.

In Section 3.4, we aim to explain that $^3DMB*$ is a more selective oxidant that prefers to react with electron-rich species such as phenols (Walling and Gibian, 1965), whereas •OH reacts fast with a wide range of organic compounds in the aqueous phase at nearly diffusion-controlled rates (Herrmann et al., 2010). Although GA reacts fast with both $^3DMB*$ and •OH, some electron-poor aqSOA products may be less reactive with $^3DMB*$ than with •OH. Previous studies on the oxidation of green leaf volatiles also suggest a higher selectivity of $^3DMB*$ than •OH. For example, the aqueous second-order rate constants for the oxidation of the green leaf volatiles by $^3DMB*$ at pH 5 and 298 K are in the range of $0.13–15 \times 10^8$ $M^{-1}$ $s^{-1}$, with the most reactive cis-3-hexenyl acetate (HxAc) being 115 times more reactive than the least reactive 2-methyl-3-butene-2-ol

(MBO) (Richards-Henderson et al., 2015). However, under the same conditions, the aqueous second-order rate constants for these green leaf volatiles with •OH are in the range of 5.3–8.6 $\times$ $10^9 \, M^{-1} \, s^{-1}$, with the most reactive HxAc being only 1.6 times more reactive than the least reactive cis-3-hexen-1-ol (HxO) (Richards-Henderson et al., 2014).

To improve the clarity of our interpretation, the text in the manuscript has been updated as follows.

Line 401: This result may reflect the reactivity differences between GA and the aqSOA towards the oxidants, i.e., while both oxidants react quickly with GA, for the electron-poor aqSOA products, $^3$C* may be much less reactive whereas •OH can oxidize them rapidly.

9. Section 3.4: Could the authors give some examples of these unique low-volatility, light-absorbing products that cannot be generated via •OH oxidation?

In this context, we are talking about the products generated from the photodegradation of DMB. According to previous studies, the photo decay of benzaldehydes can lead to the formation of dimeric products and quinones that possess highly conjugated structures and can efficiently absorb UV-vis light (Theodoropoulou et al., 2020). Scheme S1 has been added to the supplementary materials to illustrate the potential product that may arise from the photodegradation of DMB.

Scheme S1. Postulated reaction pathways for the photodegradation of 3,4-dimethoxybenzaldehyde. The mechanisms are adapted from previous studies on benzaldehydes (Berger et al., 1973; Dubtsov et al., 2006; Shen and Fang, 2011; Theodoropoulou et al., 2020).

10. Section 3.4: What do the authors mean by •OH only accounts for a small fraction of the total oxidant amount in the 3C*-initiated reaction system?

In the $^3C^*$-initiated experiment, the estimated concentration of $^3C^*$ produced by 5 µM DMB is approximately $10^{-13}$ M, while the estimated concentration of •OH produced by the addition of 100 µM $H_2O_2$ is approximately $10^{-14}$ M, which is 10 times lower than the $^3C^*$ concentration.

Minor comments and questions:

11. Intro: Is hydroxylation an example of functionalization?

Yes. Functionalization includes hydroxylation. The text in the manuscript has been updated as follows.

Line 44: The mass yields of phenolic aqSOA range from 50% to 140%, and the proposed formation pathways include oligomerization, functionalization (e.g., hydroxylation) and fragmentation (Arciva et al., 2022; Huang et al., 2018; Jiang et al., 2021; Ma et al., 2021; Smith et al., 2014, 2015, 2016; Sun et al., 2010; Yu et al., 2014, 2016).

12. Please provide more information on the HPLC method used to determine the concentration of GA and DMB.

The detection method for GA and DMB has been added to the manuscript in Sec 2.2.

Line 124: The concentrations of GA and 3,4-DMB were determined by a high-performance liquid chromatograph equipped with a diode array detector (HPLC-DAD, Agilent Technologies Inc.). A ZORBAX Eclipse XDB-$C_{18}$ column (150×4.6 mm, 5 µm) was used with a mobile phase consisting of a mixture of acetonitrile and water (20:80), and the flow rate was 0.7 mL min$^{-1}$. Both GA and DMB were detected at 280 nm, and their retention times were 8.349 and 15.396 min, respectively.

13. Please correct what [Org]t and [GA]t refer to.

Done.

14. There are other recently published articles regarding the formation of aqSOA by 3C* chemistry and the corresponding light absorption by reaction products (e.g., Li et al., 2022, https://doi.org/10.5194/acp-22-7793-2022; Mabato et al., 2023, https://doi.org/10.5194/acp-23-2859-2023)

Thanks for this comment. We have now added citations to these articles in the revised manuscript.

**Reference**

Anastasio, C., Faust, B. C. and Rao, C. J.: Aromatic Carbonyl Compounds as Aqueous-Phase Photochemical Sources of Hydrogen Peroxide in Acidic Sulfate Aerosols, Fogs, and Clouds. 1. Non-Phenolic Methoxybenzaldehydes and Methoxyacetophenones with Reductants (Phenols), Environ. Sci. Technol., 31(1), 218–232, doi:10.1021/es960359g, 1997.

Arciva, S., Niedek, C., Mavis, C., Yoon, M., Sanchez, M. E., Zhang, Q. and Anastasio, C.: Aqueous ·OH Oxidation of Highly Substituted Phenols as a Source of Secondary Organic Aerosol, Environ. Sci. Technol., 56(14), 9959–9967, doi:10.1021/acs.est.2c02225, 2022.

Berger, M., Goldblatt, I. L. and Steel, C.: Photochemistry of benzaldehyde, J. Am. Chem. Soc., 95(6), 1717–1725, doi:10.1021/ja00787a004, 1973.

Dubtsov, S. N., Dultseva, G. G., Dultsev, E. N. and Skubnevskaya, G. I.: Investigation of Aerosol Formation During Benzaldehyde Photolysis, J. Phys. Chem. B, 110(1), 645–649, doi:10.1021/jp0555394, 2006.

Herrmann, H., Hoffmann, D., Schaefer, T., Bräuer, P. and Tilgner, A.: Tropospheric Aqueous-Phase Free-Radical Chemistry: Radical Sources, Spectra, Reaction Kinetics and Prediction Tools, ChemPhysChem, 11(18), 3796–3822, doi:https://doi.org/10.1002/cphc.201000533, 2010.

Huang, D. D., Zhang, Q., Cheung, H. H. Y., Yu, L., Zhou, S., Anastasio, C., Smith, J. D. and Chan, C. K.: Formation and Evolution of aqSOA from Aqueous-Phase Reactions of Phenolic Carbonyls: Comparison between Ammonium Sulfate and Ammonium Nitrate Solutions, Environ. Sci. Technol., doi:10.1021/acs.est.8b03441, 2018.

Jiang, W., Misovich, M. V, Hettiyadura, A. P. S., Laskin, A., McFall, A. S., Anastasio, C. and Zhang, Q.: Photosensitized Reactions of a Phenolic Carbonyl from Wood Combustion in the Aqueous Phase—Chemical Evolution and Light Absorption Properties of AqSOA, Environ. Sci. Technol., 55(8), 5199–5211, doi:10.1021/acs.est.0c07581, 2021.

Ma, L., Guzman, C., Niedek, C., Tran, T., Zhang, Q. and Anastasio, C.: Kinetics and Mass Yields of Aqueous Secondary Organic Aerosol from Highly Substituted Phenols Reacting with a Triplet Excited State, Environ. Sci. Technol., 55(9), 5772–5781, doi:10.1021/acs.est.1c00575, 2021.

Richards-Henderson, N. K., Hansel, A. K., Valsaraj, K. T. and Anastasio, C.: Aqueous oxidation of green leaf volatiles by hydroxyl radical as a source of SOA: Kinetics and SOA yields, Atmos. Environ., 95, 105–112, doi:https://doi.org/10.1016/j.atmosenv.2014.06.026, 2014.

Richards-Henderson, N. K., Pham, A. T., Kirk, B. B. and Anastasio, C.: Secondary Organic Aerosol from Aqueous Reactions of Green Leaf Volatiles with Organic Triplet Excited States and Singlet Molecular Oxygen, Environ. Sci. Technol., 49(1), 268–276, doi:10.1021/es503656m, 2015.

Shen, L. and Fang, W.-H.: The Reactivity of the 1,4-Biradical Formed by Norrish Type Reactions

of Aqueous Valerophenone: A QM/MM-Based FEP Study, J. Org. Chem., 76(3), 773–779, doi:10.1021/jo101785z, 2011.

Smith, J. D., Sio, V., Yu, L., Zhang, Q. and Anastasio, C.: Secondary Organic Aerosol Production from Aqueous Reactions of Atmospheric Phenols with an Organic Triplet Excited State, Environ. Sci. Technol., 48(2), 1049–1057, doi:10.1021/es4045715, 2014.

Smith, J. D., Kinney, H. and Anastasio, C.: Aqueous benzene-diols react with an organic triplet excited state and hydroxyl radical to form secondary organic aerosol, Phys. Chem. Chem. Phys., 17(15), 10227–10237, doi:10.1039/c4cp06095d, 2015.

Smith, J. D., Kinney, H. and Anastasio, C.: Phenolic carbonyls undergo rapid aqueous photodegradation to form low-volatility, light-absorbing products, Atmos. Environ., 126, 36–44, doi:http://dx.doi.org/10.1016/j.atmosenv.2015.11.035, 2016.

Sun, Y. L., Zhang, Q., Anastasio, C. and Sun, J.: Insights into secondary organic aerosol formed via aqueous-phase reactions of phenolic compounds based on high resolution mass spectrometry, Atmos. Chem. Phys., 10(10), 4809–4822, doi:10.5194/acp-10-4809-2010, 2010.

Theodoropoulou, M. A., Nikitas, N. F. and Kokotos, C. G.: Aldehydes as powerful initiators for photochemical transformations, Beilstein J. Org. Chem., 16, 833–857, 2020.

Walling, C. and Gibian, M. J.: Hydrogen abstraction reactions by the triplet states of Ketones1, J. Am. Chem. Soc., 87(15), 3361–3364, 1965.

Yu, L., Smith, J., Laskin, A., Anastasio, C., Laskin, J. and Zhang, Q.: Chemical characterization of SOA formed from aqueous-phase reactions of phenols with the triplet excited state of carbonyl and hydroxyl radical, Atmos. Chem. Phys., 14(24), 13801–13816, doi:10.5194/acp-14-13801-2014, 2014.

Yu, L., Smith, J., Laskin, A., George, K. M., Anastasio, C., Laskin, J., Dillner, A. M. and Zhang, Q.: Molecular transformations of phenolic SOA during photochemical aging in the aqueous phase: competition among oligomerization, functionalization, and fragmentation, Atmos. Chem. Phys., 16(7), 4511–4527, doi:10.5194/acp-16-4511-2016, 2016.

---

## Author Comment (AC2)

We thank their reviewer for the thoughtful and constructive comments, and we have revised the manuscript accordingly. Listed below are our point-to-point responses (in **blue**) to the comments (in **black**) and changes of the manuscript (in **red**).

**Responses to Reviewer 2:**

Review comments on the manuscript "Photoaging of Phenolic Secondary Organic Aerosol in the Aqueous Phase: Evolution of Chemical and Optical Properties and Effects of Oxidants" in EGUsphere. The manuscript addresses the long-term aqueous aging of aqSOA formed from the photooxidation of guaiacyl acetone (GA) by OH radicals or the photosensitizer model compound 3,4-dimethoxy-benzaldehyde (DMB) using Pyrex tube experiments and performing positive matrix factorization (PMF) analysis of combined HR-AMS and UV-vis spectral data.

Questions and remarks:

1. Could the authors elaborate more the description of the experimental and analytical method, as it is a bit vague at the moment. It is not clear to the reader if the experiment is performed in a closed system or if it is, open to the atmosphere, if oxygen is present or not? Could the authors describe the potential contribution of singlet oxygen to the conversion within the system when the photosensitizer dimethoxybenzaldehyde and oxygen are present?

The Pyrex tubes containing the reaction solutions were capped but not airtight during the experiment. To ensure homogeneity, the solution was stirred continuously. In addition, the cap was briefly removed when samples were collected for instrumental analyses. Thus, oxygen should have been present in the solution throughout the reaction, allowing singlet oxygen ($^1O_2^*$) to form via energy transfer from $^3C^*$ to ground state $O_2$ and act as an oxidant for the phenolic precursor.

However, in our previous studies on the reactions of phenols with $^3DMB^*$ under similar conditions, $^1O_2^*$ has been shown to be an insignificant oxidant for the phenols compared to the rapid $^3DMB^*$-mediated reactions (Smith et al., 2014). In that work, we observed that phenols decay faster under acidic conditions, which is opposite to the fact that the reactions of $^1O_2^*$ with deprotonated phenolate ions are much more rapid (Canonica et al., 1995; Tratnyek and Hoigne, 1991). This observation suggests that $^3C^*$ is the major oxidant for phenols in the reaction, while $^1O_2^*$ is a minor contributor.

In response to these comments, we have added the following sentences to Sec 2.1 of the updated manuscript.

Line 114: During the photoreaction, the solutions were continuously stirred. The Pyrex tubes were capped but not hermetically sealed, and the caps were briefly removed during sample collection. Due to the presence of oxygen in the reaction system, secondary reactive oxygen species (ROS) such as singlet oxygen ($^1O_2^*$), superoxide/hydroperoxyl radicals ($O_2^{\bullet-}/HO_2^\bullet$) and $\bullet OH$ can be generated in the solution via energy transfer from $^3C^*$ to dissolved $O_2$ (Vione et al., 2014; Zepp et al., 1977), electron transfer from the intramolecular charge-transfer complex of DMB ($DMB^{\bullet+/\bullet-}$) to $O_2$ (Dalrymple et al., 2010; Li et al., 2022), and the reactions between DMB ketyl radical and

O₂ (Anastasio et al., 1997). However, according to our previous studies (Smith et al., 2014), singlet oxygen is expected to contribute only minimally to the oxidation of GA in this reaction system. In addition, the negligible loss of GA and DMB in the dark controls suggests there was negligible evaporation of the precursor or the photosensitizer during the experiments.

2. Could the guaiacylacetone itself act as a photosensitizer?

Compared with DMB, GA absorbs much less sunlight due to its lower extent of conjugation. This suggests that GA is not an effective photosensitizer, and the formation of triplet excited state of GA is negligible in the reaction. In a study by Smith et al. (2016), it was reported that GA is significantly less light-absorbing compared to other phenolic and non-phenolic carbonyl photosensitizers, and shows negligible direct photolysis or oxidation with syringol. To show the light absorptivity difference between GA and DMB, we have updated Figure S5 to include the MAC spectra of both compounds.

[Figure]

Figure S5. Mass absorption coefficient (MAC) spectra of guaiacyl acetone (GA) and 3,4-dimethoxybenzaldehyde (DMB)

3. Would the authors expect a similar oligomer formation yield and conversion rate if 4-propylguaiacol had been used?

We would expect oligomer formation from the photoreaction of 4-propylguaiacol with ³C*/•OH since it is a methoxyphenol. However, different phenols can exhibit significantly different oligomer formation rate and yields. For example, Yu et al. (2014) reported that the aqSOA formed from guaiacol shows a significantly higher mass fraction of oligomers than that from phenol. In addition, the formation rates of oligomers from the coupling of phenoxyl radicals can be affected by steric hindrance by large functional groups (Steenken and Neta, 2003).

Moreover, it is important to note that GA has a ketone functional group which is absent in 4-propylguaiacol, and the reactions triggered by the ketone group can also contribute the production of oligomers as shown in Scheme 2 in the updated manuscript.

Scheme 2. Postulated reaction pathways triggered by the ketone functional group of GA.

4. What would be the general reaction mechanism in the photosensitizer system in the presence of GA and DMB after the first step of H-atom abstraction or electron transfer? Is addition of the photosensitizer possible?

The reaction of $^3DMB^*$ with GA can lead to H-atom abstraction/electron transfer at the phenol functional group, generating a GA phenoxyl radical and a DMB ketyl radical. The GA phenoxyl radicals can undergo coupling to form dimers and higher oligomers (Mabato et al., 2023; Yu et al., 2014). Alternatively, the GA phenoxyl radical can also react with $HO_2\bullet/\bullet O_2^-$ in the solution to produce o-quinone, which is subsequently converted to hydroxylated GA products (D'Alessandro et al., 2000; Steenken and Neta, 2003). The DMB ketyl radical can be reduced by $O_2$ to form $HO_2\bullet/\bullet O_2^-$ and regenerate DMB (Anastasio et al., 1997). Scheme 1 has been added to the updated manuscript to illustrate the reaction pathways.

[Figure]

Scheme 1. Postulated reaction pathways triggered by the phenol functional group of GA.

On the other hand, during the photoexcitation, DMB may dissociate via Norrish reactions to generate phenyl and benzoyl radicals (Shen and Fang, 2011) which can undergo recombination to produce dimeric products and dimethoxybenzene (Dubtsov et al., 2006; Theodoropoulou et al., 2020). Therefore, as this reviewer suggested, DMB may participate in the reactions as a reactant. Scheme S1 has been added to the supplementary to illustrate potential DMB-participated reaction pathways.

However, considering the low concentration of DMB (20 times lower than GA) and its slow decay rate observed in our reaction system, it is unlikely that the reactions involving DMB played a significant role, and their contribution to the aqSOA is expected to be insignificant.

Scheme S1. Postulated reaction pathways for the photodegradation of 3,4-dimethoxybenzaldehyde. The mechanisms are adapted from previous studies on benzaldehydes (Berger et al., 1973; Dubtsov et al., 2006; Shen and Fang, 2011; Theodoropoulou et al., 2020).

5. What would be the difference in the first oxidation products formed by the photosensitizers compared to OH radicals?

It is important to note that based on our AMS measurements, the aqSOA was found to be a mixture of different products from the beginning, reflecting the competition of different reaction pathways (e.g., oligomerization, functionalization, and fragmentation) throughout the aqSOA formation and aging. The postulated reaction pathways are shown in Schemes 1 and 2.

Based on our PMF analysis, both the 1st-generation aqSOA factor (which is enrich with oligomers) and the 2nd-generation factor (which is enriched with functionalized GA monomers) were found to build up upon irradiation in both the •OH and the $^3$C* reactions. The mass spectral profiles of the 1st-generation aqSOA factors from the $^3$C* and •OH reactions were found to be highly similar, suggesting similar compositions. However, the mass fraction of the 1st-generation factor in the $^3$C*-aqSOA is higher than that in the •OH-aqSOA initially. For example, when 20% of GA was consumed, the 1st-generation factor accounted for ~ 75% of the total •OH-aqSOA mass (Figure 6e), while it constituted 90% of the total $^3$C*-aqSOA mass (Figure 7e). This suggests that oligomerization plays a more important role in the early stage of $^3$C*-mediated reactions as compared to •OH reactions. A more detailed comparison between the initially formed $^3$C*-aqSOA and •OH-aqSOA is shown in Sec 3.1 of the manuscript.

6. What would be the result of the involvement of oxygen? Would the resulting peroxyl radicals lead to the formation of oligomers, what is the authors' opinion? If not, could this explain the lower yield of oligomers in OH radical-induced oxidation? How likely is alkyl/phenoxy-like radical recombination in the presence of oxygen at the steady-state concentrations used?

The presence of oxygen may influence the reactions in several ways, as described below.

1) In the •OH-mediated reaction, the dihydroxycyclohexadienyl radical generated via OH addition can further react with $O_2$ to form a peroxyl radical (Scheme 1). The peroxyl radical can eliminate a $HO_2$• to produce hydroxylated products. Alternatively, the peroxyl radical can also cyclize to generate bicyclic radicals (Dong et al., 2021; Suh et al., 2003), which can undergo ring cleavage pathways to produce fragmented products such as small carboxylic acids, aldehydes, and ketones.

However, we do not expect dimerization of the peroxyl radicals to be significant in our reaction system since we did not observe the formation of organic peroxide (ROOR). Oligomer formation in this study is likely primarily through the coupling of phenoxyl radicals, which is more enhanced in the $^3C^*$-mediated reaction. Therefore, we agree with this reviewer that negligible dimerization of peroxyl radical may also partly explain the lower yield of oligomers in the •OH-mediated reaction.

In summary, the presence of $O_2$ in the •OH reaction can enhance ring-cleavage reactions, while the oligomerization of phenoxyl radicals can be suppressed (Dong et al., 2021). However, we cannot determine the degree to which the presence of $O_2$ inhibits oligomerization.

2) In the $^3C^*$-mediated reaction, the energy transfer from $^3DMB^*$ to ground state $O_2$ produces singlet oxygen ($^1O_2^*$; Scheme 1). The $^1O_2^*$ reacts with phenols mainly through 1,4-cycloaddition route to produce quinoic products (Al-Nu'airat et al., 2019; García, 1994). A higher $O_2$ concentration increases the importance of $^1O_2^*$ pathways, thus promoting quinone formation. However, it also suppresses the formation of phenoxyl radicals and the dimerization of phenoxyl radicals.

However, as discussed in our response to the first Question of this reviewer, the $^1O_2^*$ reaction is expected to be a minor contributor to GA oxidation in this study.

3) Furthermore, in the $^3C^*$ reaction, DMB ketyl radical can react with $O_2$ to produce superoxide/hydroperoxyl radical ($O_2$•$^-$/$HO_2$•) which can further react to form $H_2O_2$ and •OH (Scheme 1). The $O_2$•$^-$/$HO_2$• and •OH play an important role in the hydroxylation and ring-opening reactions of GA. Therefore, the presence of $O_2$ in the $^3C^*$ reaction can contribute to more promoted hydroxylation and ring-opening pathways.

4) The addition of ground state $O_2$ to the phenoxyl radical is possible, which leads to the formation of a phenoxy peroxy radical (Batiha et al., 2012; Steenken and Neta, 2003). However, this reaction is very slow and does not appear to significantly affect the fate of phenoxyl radicals (Batiha et al., 2012). Therefore, the effect of this pathway on the dimerization of phenoxyl radicals is expected to be negligible.

To demonstrate the aqueous-phase reaction pathways of GA and the role of $O_2$ in the reactions, we have added "Section 3.2 Aqueous-phase Reaction Pathways of Guaiacyl Acetone" and Schemes 1 and 2 to the updated manuscript.

7. In the authors' opinion, what are the main oxidation products of the OH radical reaction with GA?

The composition of the aqSOA evolves over time, with oligomers and functionalized products being more important in the early stages of the reaction. However, as the aging process continues, ring-opening products (e.g., small acids, aldehydes and ketones) become more dominant. These findings are demonstrated in Figures 6e, 6g, and 6i, and are consistent with the proposed reaction mechanisms. We have added Schemes 1 and 2 to the updated manuscript to illustrate the major reaction pathways and the formation of these products.

8. On page 7, line 166, it is mentioned that the products formed can evaporate more easily. Later (line 177), the formation of carboxylic acids and compounds formed by the degradation of aromatic rings is mentioned. How likely is it that these compounds will evaporate?

In this study, we did not characterize the volatility of the products. However, our previous study (Jiang et al., 2021) observed the formation of five ring-opening carboxylic acids, including formic acid, acetic acid, malonic acid, malic acid, and oxalic acid, from the aqueous photoreactions of GA. As shown in the following table, most of the detected acids are volatile and are expected to evaporate readily from the aerosol phase.

| Name | Formula | $K_H$ (M atm$^{-1}$) (EPI Suite, HENRYWIN) | Vapor Pressure at 298 K (Pa) (EPI Suite, MPBPWIN) | Fraction sorbed to airbone particles (EPI Suite, AEROWIN, Mackay Model) |
|---|---|---|---|---|
| Formic Acid | HCHO | 1.96E+03 | 4.78E+03 | 4.23E-08 |
| Acetic Acid | C2H4O2 | 3.40E+03 | 2.29E+03 | 1.15E-07 |
| Malonic Acid | C3H4O4 | 2.46E+08 | 1.34E-01 | 9.78E-05 |
| Malic Acid | C4H6O5 | 1.19E+09 | 3.90E-04 | 8.31E-01 |
| Oxalic Acid | C2H2O4 | 4.15E+07 | 7.46E-01 | 1.34E-03 |

9. Could the authors indicate how long the H2O2 (100 uM) as well as the DMB is present in the solution before it decays by the photochemistry? How many times is DMB involved in a reaction as a photosensitizer before it is degraded?

Based on the fact that GA follows pseudo-first-order decay until completely consumed in both the •OH and the $^3$C* reactions, we deduce that the concentrations of $H_2O_2$ and DMB stayed relatively constant during the initial aqSOA formation period (i.e., 24 hours for $H_2O_2$ and 3.5 hours for DMB). After GA is consumed, we were not able to monitor the concentration of $H_2O_2$ but we measured the concentration changes of DMB using HPLC-DAD and the results are shown in Figure 1i.

The times that a DMB molecule can be involved in the reactions as a photosensitizer are influenced by several factors, such as the rate of DMB photoexcitation, the quantum yield of $^3$DMB*, the rate of H-atom abstraction/electron transfer between $^3$DMB* and GA, other potential sinks for $^3$DMB* (e.g., $O_2$) in the system, and the rate of DMB photo-dissociation. Although this is an interesting question, it is beyond the scope of this work, which focuses on understanding the chemical and optical properties of the GA aqSOA.

10. Is it possible that GA acts as a photosensitizer in the GA + H2O2 system used and reacts with H2O2, which is subsequently more important than the production of OH radicals by H2O2 photolysis, since the absorption of GA is somewhat greater compared to H2O2 in the specific wavelength range?

Based on our previous study, we do not expect GA to be an efficient photosensitizer, as both direct photolysis of GA and $^3C^*$ formation from GA (monitored by the reaction between syringol and GA) were found to be negligible under a similar condition (Smith et al., 2016).

11. Did the authors do any experiments with GA in the absence of an oxidant?

Our previous study has reported that GA exhibits negligible direct photolysis under simulated solar irradiation (Smith et al., 2016). The following sentence has been added to the manuscript.

Line 162: Based on a previous study (Smith et al., 2016), direct photolysis of GA is expected to be negligible in this study.

12. How justified here is the statement (on page 10 line 215) that the steady-state concentration of OH is the same or similar?

We assumed a steady-state concentration of •OH during the initial 24 hours of illumination based on the fact that GA followed pseudo-first-order decay during that time period. However, as GA was depleted after 24 h of irradiation, we were unable to predict •OH concentration based on the decay of GA. It is possible that the concentration of •OH decreased due to the consumption of $H_2O_2$, especially towards the end of the prolonged aging (72 hr).

13. The end of section 3.4 is difficult to follow and is full of speculation. Could the authors sharpen the end of the section for clarity and with concrete numbers, e.g., how likely evaporation is? This brings me back to the experimental description, where it is not clear whether evaporation may or may not play a role in this study, so the experimental design is not well described within the manuscript.

We would like to point out that AMS only measures the low volatility products, while the volatile and semi-volatile products evaporated during the aerosolization and drying. For clarification, the following sentences have been added to Sec 2.2.

Line 129: The liquid samples were atomized in argon (Ar, industrial grade, 99.997 %) followed by diffusion drying (Jiang et al., 2021). This process allowed volatile and semi-volatile products to evaporate, leaving only the low-volatility products in the particle phase, which were characterized by AMS.

In addition, the following text has been added to Sec 2.1 to better describe the experimental method.

Line 114: During the photoreaction, the solutions were continuously stirred. The Pyrex tubes were capped but not hermetically sealed, and the caps were briefly removed during sample collection. Due to the presence of oxygen in the reaction system, secondary reactive oxygen species (ROS) such as singlet oxygen ($^1O_2^*$), superoxide/hydroperoxyl radicals ($O_2^{•-}/HO_2^•$) and •OH can be generated in the solution via energy transfer from $^3C^*$ to dissolved $O_2$ (Vione et al., 2014; Zepp et

al., 1977), electron transfer from the intramolecular charge-transfer complex of DMB (DMB$\bullet^{+/\bullet-}$) to $O_2$ (Dalrymple et al., 2010; Li et al., 2022), and the reactions between DMB ketyl radical and $O_2$ (Anastasio et al., 1997). However, according to our previous studies (Smith et al., 2014), singlet oxygen is expected to contribute only minimally to the oxidation of GA in this reaction system. In addition, the negligible loss of GA and DMB in the dark controls suggests there was negligible evaporation of the precursor or the photosensitizer during the experiments.

To better interpret the effects of additional oxidants on the aqSOA aging, the following text has been added to Sec 3.4.

Line 386: Likewise, the addition of extra $\bullet$OH or $^3$C* results in more extensive mass loss of the $\bullet$OH-aqSOA, with reductions of 88% or 79% of the aqSOA mass observed at the end of the photoaging, respectively. These levels of mass loss were significantly higher than no extra oxidant (i.e., 62%). These findings suggest that the presence of additional $\bullet$OH or $^3$C* accelerates the photochemical aging process and leads to increased formation of volatile and semi-volatile products that subsequently evaporate.

In summary, this manuscript, which certainly has its merits and is quite interesting in its present form, might be improved more in terms of the clarity and coherence of its scientific basis to enable a recommendation for acceptance.

We have carefully considered this comment, as well as all the other comments, and have revised our manuscript accordingly. Specifically, we have made changes to address the concerns raised by the reviewer, including modifying the language in certain sections and providing additional clarification and explanation where needed. We have also added new data and reaction schematics to support our findings. We believe that these revisions have strengthened our manuscript and improved its overall quality.

**Reference**

Al-Nu'airat, J., Dlugogorski, B. Z., Gao, X., Zeinali, N., Skut, J., Westmoreland, P. R., Oluwoye, I. and Altarawneh, M.: Reaction of phenol with singlet oxygen, Phys. Chem. Chem. Phys., 21(1), 171–183, doi:10.1039/C8CP04852E, 2019.

Anastasio, C., Faust, B. C. and Rao, C. J.: Aromatic Carbonyl Compounds as Aqueous-Phase Photochemical Sources of Hydrogen Peroxide in Acidic Sulfate Aerosols, Fogs, and Clouds. 1. Non-Phenolic Methoxybenzaldehydes and Methoxyacetophenones with Reductants (Phenols), Environ. Sci. Technol., 31(1), 218–232, doi:10.1021/es960359g, 1997.

Batiha, M., Al-Muhtaseb, A. H. and Altarawneh, M.: Theoretical study on the reaction of the phenoxy radical with O2, OH, and NO2, Int. J. Quantum Chem., 112(3), 848–857, doi:10.1002/qua.23074, 2012.

Berger, M., Goldblatt, I. L. and Steel, C.: Photochemistry of benzaldehyde, J. Am. Chem. Soc., 95(6), 1717–1725, doi:10.1021/ja00787a004, 1973.

Canonica, S., Jans, U. R. S., Stemmler, K. and Hoigne, J.: Transformation kinetics of phenols in water: photosensitization by dissolved natural organic material and aromatic ketones, Environ. Sci. Technol., 29(7), 1822–1831, 1995.

D'Alessandro, N., Bianchi, G., Fang, X., Jin, F., Schuchmann, H.-P. and von Sonntag, C.: Reaction of superoxide with phenoxyl-type radicals, J. Chem. Soc. Perkin Trans. 2, (9), 1862–1867, doi:10.1039/B003346O, 2000.

Dalrymple, R. M., Carfagno, A. K. and Sharpless, C. M.: Correlations between Dissolved Organic Matter Optical Properties and Quantum Yields of Singlet Oxygen and Hydrogen Peroxide, Environ. Sci. Technol., 44(15), 5824–5829, doi:10.1021/es101005u, 2010.

Dong, P., Chen, Z., Qin, X. and Gong, Y.: Water Significantly Changes the Ring-Cleavage Process During Aqueous Photooxidation of Toluene, Environ. Sci. Technol., 55(24), 16316–16325, doi:10.1021/acs.est.1c04770, 2021.

Dubtsov, S. N., Dultseva, G. G., Dultsev, E. N. and Skubnevskaya, G. I.: Investigation of Aerosol Formation During Benzaldehyde Photolysis, J. Phys. Chem. B, 110(1), 645–649, doi:10.1021/jp0555394, 2006.

García, N. A.: New trends in photobiology: Singlet-molecular-oxygen-mediated photodegradation of aquatic phenolic pollutants. A kinetic and mechanistic overview, J. Photochem. Photobiol. B Biol., 22(3), 185–196, doi:https://doi.org/10.1016/1011-1344(93)06932-S, 1994.

Jiang, W., Misovich, M. V, Hettiyadura, A. P. S., Laskin, A., McFall, A. S., Anastasio, C. and Zhang, Q.: Photosensitized Reactions of a Phenolic Carbonyl from Wood Combustion in the Aqueous Phase—Chemical Evolution and Light Absorption Properties of AqSOA, Environ. Sci. Technol., 55(8), 5199–5211, doi:10.1021/acs.est.0c07581, 2021.

Li, X., Tao, Y., Zhu, L., Ma, S., Luo, S., Zhao, Z., Sun, N., Ge, X. and Ye, Z.: Optical and chemical properties and oxidative potential of aqueous-phase products from OH and 3C∗-initiated photooxidation of eugenol, Atmos. Chem. Phys., 22(11), 7793–7814, doi:10.5194/acp-22-7793-2022, 2022.

Mabato, B. R. G., Li, Y. J., Huang, D. D., Wang, Y. and Chan, C. K.: Comparison of aqueous secondary organic aerosol (aqSOA) product distributions from guaiacol oxidation by non-phenolic and phenolic methoxybenzaldehydes as photosensitizers in the absence and presence of ammonium nitrate, Atmos. Chem. Phys., 23(4), 2859–2875, doi:10.5194/acp-23-2859-2023, 2023.

Shen, L. and Fang, W.-H.: The Reactivity of the 1,4-Biradical Formed by Norrish Type Reactions of Aqueous Valerophenone: A QM/MM-Based FEP Study, J. Org. Chem., 76(3), 773–779, doi:10.1021/jo101785z, 2011.

Smith, J. D., Sio, V., Yu, L., Zhang, Q. and Anastasio, C.: Secondary Organic Aerosol Production from Aqueous Reactions of Atmospheric Phenols with an Organic Triplet Excited State, Environ. Sci. Technol., 48(2), 1049–1057, doi:10.1021/es4045715, 2014.

Smith, J. D., Kinney, H. and Anastasio, C.: Phenolic carbonyls undergo rapid aqueous photodegradation to form low-volatility, light-absorbing products, Atmos. Environ., 126, 36–44, doi:http://dx.doi.org/10.1016/j.atmosenv.2015.11.035, 2016.

Steenken, S. and Neta, P.: Transient Phenoxyl Radicals: Formation and Properties in Aqueous Solutions, in The Chemistry of Phenols, pp. 1107–1152., 2003.

Suh, I., Zhang, R., Molina, L. T. and Molina, M. J.: Oxidation Mechanism of Aromatic Peroxy and Bicyclic Radicals from OH−Toluene Reactions, J. Am. Chem. Soc., 125(41), 12655–12665, doi:10.1021/ja0350280, 2003.

Theodoropoulou, M. A., Nikitas, N. F. and Kokotos, C. G.: Aldehydes as powerful initiators for photochemical transformations, Beilstein J. Org. Chem., 16, 833–857, 2020.

Tratnyek, P. G. and Hoigne, J.: Oxidation of substituted phenols in the environment: a QSAR analysis of rate constants for reaction with singlet oxygen, Environ. Sci. Technol., 25(9), 1596–1604, 1991.

Vione, D., Minella, M., Maurino, V. and Minero, C.: Indirect Photochemistry in Sunlit Surface Waters: Photoinduced Production of Reactive Transient Species, Chem. – A Eur. J., 20(34), 10590–10606, doi:https://doi.org/10.1002/chem.201400413, 2014.

Yu, L., Smith, J., Laskin, A., Anastasio, C., Laskin, J. and Zhang, Q.: Chemical characterization of SOA formed from aqueous-phase reactions of phenols with the triplet excited state of carbonyl and hydroxyl radical, Atmos. Chem. Phys., 14(24), 13801–13816, doi:10.5194/acp-14-13801-2014, 2014.

Zepp, R. G., Wolfe, N. L., Baughman, G. L. and Hollis, R. C.: Singlet oxygen in natural waters, Nature, 267(5610), 421–423, doi:10.1038/267421a0, 1977.

---

## Author Comment (AC3)

[revised manuscript text omitted]

The light absorption coefficient ($\alpha_\lambda$, cm$^{-1}$) of the aqSOA was calculated as:

$$\alpha_\lambda = \frac{A_{total,\lambda} - A_{GA,\lambda} - A_{DMB,\lambda}}{l} \tag{Eq. S1}$$

where $A_{total,\lambda}$ is the total measured base-10 light absorbance of the solution at wavelength $\lambda$, $A_{GA,\lambda}$ and $A_{DMB,\lambda}$ denote the absorbance contributed by GA and 3,4-DMB, and l is the pathlength of the cuvette (1 cm). The mass absorption coefficient ($MAC_\lambda$, m$^2$ g$^{-1}$) of the aqSOA was calculated as:

$$MAC_\lambda = \frac{2.303 \times \alpha_\lambda}{[Org]_{solution}} \times 100 \tag{Eq. S2}$$

where $[Org]_{solution}$ is the aqSOA mass concentration ($\mu$g mL$^{-1}$) in the solution, 2.303 is a conversion factor between log10 and natural log, and 100 is for unit conversion. The absorption Ångström exponent (AAE) of the aqSOA was calculated as:

$$AAE_{\lambda1-\lambda2} = -\frac{\ln\frac{\alpha_{\lambda1}}{\alpha_{\lambda2}}}{\ln\frac{\lambda1}{\lambda2}} \tag{Eq. S3}$$

where $\alpha_{\lambda1}$ and $\alpha_{\lambda2}$ denote the light absorption coefficients at wavelengths $\lambda_1$ and $\lambda_2$. The rate of sunlight absorption of the aqSOA ($R_{abs}$, mol photons L$^{-1}$ s$^{-1}$) was calculated as:

$$R_{abs} = 2.303 \times \frac{10^3}{N_A} \times \sum_{290\ nm}^{500\ nm}(\alpha_\lambda \times I_\lambda \times \Delta\lambda) \tag{Eq. S4}$$

where $I_\lambda$ is the midday winter-solstice actinic flux in Davis (photons cm$^{-2}$ s$^{-1}$ nm$^{-1}$) from the Tropospheric Ultraviolet and Visible (TUV) Radiation Model version 5.3 (https://www.acom.ucar.edu/Models/TUV/Interactive_TUV/), $\Delta\lambda$ is the interval between adjacent wavelengths in the TUV output, 2.303 is for base conversion between log10 and natural log, $10^3$ is for unit conversion, and $N_A$ is Avogadro's number.

**Table S1. The •OH-aqSOA mass yield, H/C, O/C and OS$_C$ determined by HR-ToF-AMS during aqSOA formation and aging.**

| | Irradiation Time (h) | SOA yield | H/C | O/C | OSc |
|---|---|---|---|---|---|
| •OH-aqSOA formation | 0.5 | 9.72E-01 | 1.64 | 0.32 | -1.00 |
| | 1 | 8.60E-01 | 1.52 | 0.43 | -0.66 |
| | 2 | 1.03E+00 | 1.46 | 0.51 | -0.45 |
| | 3 | 1.06E+00 | 1.43 | 0.54 | -0.34 |
| | 4 | 9.24E-01 | 1.43 | 0.55 | -0.33 |
| | 6 | 8.16E-01 | 1.42 | 0.57 | -0.27 |
| | 8 | 7.39E-01 | 1.41 | 0.60 | -0.21 |
| | 10 | 6.70E-01 | 1.41 | 0.60 | -0.21 |
| | 12 | 5.93E-01 | 1.39 | 0.62 | -0.15 |
| | 24 | 4.60E-01 | 1.38 | 0.65 | -0.10 |
| •OH-aqSOA aging (no addition of extra oxidant) | 25 | 4.41E-01 | 1.38 | 0.66 | -0.06 |
| | 26 | 4.31E-01 | 1.37 | 0.66 | -0.06 |
| | 28 | 4.16E-01 | 1.37 | 0.66 | -0.06 |
| | 30 | 3.96E-01 | 1.37 | 0.67 | -0.03 |
| | 35 | 3.55E-01 | 1.36 | 0.67 | -0.02 |
| | 36 | 3.50E-01 | 1.36 | 0.67 | -0.02 |
| | 40 | 3.26E-01 | 1.36 | 0.67 | -0.02 |
| | 44 | 3.05E-01 | 1.36 | 0.67 | -0.01 |
| | 48 | 2.82E-01 | 1.36 | 0.67 | -0.02 |
| | 60 | 2.34E-01 | 1.35 | 0.67 | -0.01 |
| | 70 | 2.07E-01 | 1.36 | 0.67 | -0.01 |
| | 72 | 2.02E-01 | 1.35 | 0.68 | 0.02 |
| •OH-aqSOA aging (add 100 µM of H2O2) | 25 | 4.54E-01 | 1.39 | 0.68 | -0.03 |
| | 26 | 4.48E-01 | 1.39 | 0.70 | 0.01 |
| | 28 | 3.74E-01 | 1.39 | 0.70 | 0.01 |
| | 30 | 3.10E-01 | 1.39 | 0.70 | 0.01 |
| | 35 | 1.96E-01 | 1.40 | 0.68 | -0.04 |
| | 36 | 1.85E-01 | 1.41 | 0.66 | -0.08 |
| | 40 | 1.56E-01 | 1.41 | 0.64 | -0.13 |
| | 44 | 1.21E-01 | 1.40 | 0.64 | -0.12 |
| | 48 | 9.55E-02 | 1.40 | 0.64 | -0.12 |
| | 60 | 7.43E-02 | 1.43 | 0.61 | -0.21 |
| | 70 | 6.39E-02 | 1.44 | 0.57 | -0.29 |
| | 72 | 6.14E-02 | 1.44 | 0.57 | -0.29 |
| •OH-aqSOA aging (add 5 µM of 3,4-DMB) | 25 | 4.88E-01 | 1.38 | 0.66 | -0.05 |
| | 26 | 4.48E-01 | 1.38 | 0.67 | -0.04 |
| | 28 | 4.20E-01 | 1.37 | 0.69 | 0.01 |
| | 30 | 3.99E-01 | 1.39 | 0.68 | -0.03 |
| | 35 | 2.99E-01 | 1.40 | 0.67 | -0.05 |
| | 36 | 2.85E-01 | 1.40 | 0.67 | -0.05 |
| | 40 | 2.40E-01 | 1.40 | 0.66 | -0.08 |
| | 44 | 2.14E-01 | 1.41 | 0.66 | -0.09 |
| | 48 | 1.89E-01 | 1.41 | 0.65 | -0.10 |
| | 60 | 1.37E-01 | 1.41 | 0.64 | -0.12 |
| | 70 | 1.13E-01 | 1.42 | 0.62 | -0.17 |
| | 72 | 1.10E-01 | 1.43 | 0.61 | -0.20 |
| •OH-aqSOA aging (dark) | 24 | 4.29E-01 | 1.39 | 0.65 | -0.09 |
| | 48 | 4.31E-01 | 1.39 | 0.66 | -0.07 |
| | 72 | 4.43E-01 | 1.37 | 0.66 | -0.05 |

**Table S2. The $^3C^*$-aqSOA mass yield, H/C, O/C and $OS_C$ determined by HR-ToF-AMS during aqSOA formation and aging.**

|  | Irradiation Time (h) | SOA yield | H/C | O/C | OSc |
|---|---|---|---|---|---|
| $^3C^*$-aqSOA formation | 0.3 | / | 1.61 | 0.37 | -0.88 |
|  | 0.6 | 9.03E-01 | 1.53 | 0.43 | -0.67 |
|  | 0.9 | 8.90E-01 | 1.50 | 0.46 | -0.58 |
|  | 1.2 | 8.83E-01 | 1.49 | 0.48 | -0.52 |
|  | 1.7 | 8.69E-01 | 1.48 | 0.51 | -0.46 |
|  | 2.3 | 8.58E-01 | 1.47 | 0.54 | -0.39 |
|  | 2.9 | 8.50E-01 | 1.46 | 0.54 | -0.38 |
|  | 3.5 | 8.58E-01 | 1.44 | 0.58 | -0.29 |
| $^3C^*$-aqSOA aging (no addition of extra oxidant) | 3.8 | 8.39E-01 | 1.43 | 0.60 | -0.23 |
|  | 4.1 | 8.52E-01 | 1.43 | 0.62 | -0.17 |
|  | 4.3 | 8.45E-01 | 1.42 | 0.64 | -0.15 |
|  | 4.6 | 8.23E-01 | 1.43 | 0.65 | -0.13 |
|  | 4.9 | 8.13E-01 | 1.43 | 0.64 | -0.15 |
|  | 5.2 | 8.02E-01 | 1.42 | 0.68 | -0.06 |
|  | 5.8 | 7.80E-01 | 1.42 | 0.71 | -0.01 |
|  | 6.4 | 7.49E-01 | 1.42 | 0.73 | 0.03 |
|  | 7.0 | 7.16E-01 | 1.42 | 0.74 | 0.05 |
|  | 8.1 | 6.33E-01 | 1.43 | 0.75 | 0.06 |
|  | 9.3 | 5.76E-01 | 1.43 | 0.76 | 0.10 |
|  | 11.6 | 5.05E-01 | 1.43 | 0.77 | 0.11 |
|  | 13.9 | 4.43E-01 | 1.43 | 0.77 | 0.12 |
| $^3C^*$-aqSOA aging (add 100 µM of H2O2) | 3.8 | 8.44E-01 | 1.44 | 0.59 | -0.24 |
|  | 4.1 | 8.42E-01 | 1.43 | 0.62 | -0.17 |
|  | 4.3 | 8.33E-01 | 1.42 | 0.65 | -0.12 |
|  | 4.6 | 8.04E-01 | 1.43 | 0.68 | -0.06 |
|  | 4.9 | 7.95E-01 | 1.42 | 0.70 | -0.02 |
|  | 5.2 | 7.83E-01 | 1.42 | 0.70 | -0.01 |
|  | 5.8 | 7.52E-01 | 1.41 | 0.72 | 0.03 |
|  | 6.4 | 7.14E-01 | 1.41 | 0.73 | 0.05 |
|  | 7.0 | 6.85E-01 | 1.42 | 0.74 | 0.07 |
|  | 8.1 | 6.30E-01 | 1.42 | 0.75 | 0.08 |
|  | 9.3 | 5.70E-01 | 1.43 | 0.76 | 0.10 |
|  | 11.6 | 4.89E-01 | 1.45 | 0.74 | 0.02 |
|  | 13.9 | 3.81E-01 | 1.45 | 0.76 | 0.07 |
| $^3C^*$-aqSOA aging (add 5 µM of 3,4-DMB) | 3.8 | 8.48E-01 | 1.42 | 0.58 | -0.30 |
|  | 4.1 | 8.44E-01 | 1.42 | 0.59 | -0.28 |
|  | 4.3 | 8.50E-01 | 1.42 | 0.59 | -0.26 |
|  | 4.6 | 8.46E-01 | 1.42 | 0.59 | -0.22 |
|  | 4.9 | 8.41E-01 | 1.43 | 0.61 | -0.19 |
|  | 5.2 | 8.41E-01 | 1.43 | 0.61 | -0.18 |
|  | 5.8 | 8.31E-01 | 1.41 | 0.63 | -0.15 |
|  | 6.4 | 8.22E-01 | 1.42 | 0.65 | -0.12 |
|  | 7.0 | 8.08E-01 | 1.42 | 0.66 | -0.09 |
|  | 8.1 | 7.73E-01 | 1.42 | 0.68 | -0.07 |
|  | 9.3 | 7.45E-01 | 1.42 | 0.69 | -0.04 |
|  | 11.6 | 6.82E-01 | 1.43 | 0.70 | -0.02 |
|  | 13.9 | 6.38E-01 | 1.43 | 0.71 | -0.02 |
| $^3C^*$-aqSOA aging (dark) | 7.0 | 9.05E-01 | 1.44 | 0.59 | -0.26 |
|  | 13.9 | 9.25E-01 | 1.43 | 0.60 | -0.24 |

95

**Table S3. Exponential fits for aqSOA formation and decay.**

| | Pseudo-first-order decay of GA | Exponential fit for initial aqSOA formation | Exponential fit for aqSOA decay | |
|---|---|---|---|---|
| •OH-aqSOA | $[GA]_t/[GA]_0 = e^{-0.144t}$ | $y = 14.8(1-e^{-0.167x})$ | No addition of extra oxidant: | $y = 7.4e^{-0.017x}$ |
| | | | Add 100 µM of $H_2O_2$: | $y = 6.6e^{-0.11x}+1.08$ |
| | | | Add 5 µM of 3,4-DMB: | $y = 6.3e^{-0.057x}+1.44$ |
| $^3C*$-aqSOA | $[GA]_t/[GA]_0 = e^{-0.727t}$ | $y = 14.5(1-e^{-0.945x})$ | No addition of extra oxidant: | $y = 15.1e^{-0.073x}$ |
| | | | Add 100 µM of $H_2O_2$: | $y = 15.3e^{-0.078x}$ |
| | | | Add 5 µM of 3,4-DMB: | $y = 15.7e^{-0.034x}$ |

[Figure]

**Figure S1.** Summary of diagnostic plots of the PMF analysis of the •OH-initiated reactions : (a) $Q/Q_{exp}$ as a function of number of factors selected for PMF modeling. (b) $Q/Q_{exp}$ as a function of fPeak. (c) Correlations among PMF factors. (d) Box and whisker plot showing the distributions of scaled residuals for each AMS ion. (e) Box and whisker plot showing the distributions of scaled residuals for each light absorption wavelength. (f) Reconstructed and measured total signal for each sample. (g) $Q/Q_{exp}$ for each sample.

100

105

[Figure]

**Figure S2.** Summary of diagnostic plots of the PMF analysis of the $^3$C*-initiated reactions: (a) Q/Qexp as a function of number of factors selected for PMF modeling. (b) Q/Qexp as a function of fPeak. (c) Correlations among PMF factors. (d) Box and whisker plot showing the distributions of scaled residuals for each AMS ion. (e) Box and whisker plot showing the distributions of scaled residuals for each light absorption wavelength. (f) Reconstructed and measured total signal for each sample. (g) Q/Qexp for each
110 sample.

[Figure]

**Figure S3. Evolution of the mass absorption coefficient spectra of the GA •OH-aqSOA.**

115

[Figure]

**Figure S4. Evolution of the mass absorption coefficient spectra of the GA ³C*-aqSOA.**

[Figure]

120    **Figure S5. Mass absorption coefficient spectra of guaiacyl acetone and 3,4-dimethoxybenzaldehyde.**

[Figure]

**Figure S6. AMS spectra of the •OH-aqSOA and ³C\*-aqSOA before and after aging in the dark.**

[Figure]

125

**Figure S7. Van Krevelen diagrams that illustrate the evolution trends of the •OH-aqSOA and ³C\*-aqSOA under different photoaging conditions.**

[Figure]

130    **Figure S8. Triangle plots ($f_{CO2+}$ vs $f_{C2H3O+}$) that depict the evolution trends of the •OH-aqSOA and $^3$C\*-aqSOA under different photoaging conditions.**

[Figure]

**Figure S9.** The plots of $f_{CO2+}$ vs $f_{CHO2+}$ that depict the carboxylic acid formation in the •OH-aqSOA and the $^3C^*$-aqSOA under different photoaging conditions.

[Figure]

**Figure S10.** Time trend of selected AMS tracer ions in the •OH-aqSOA during aqSOA formation and prolonged aging.

[Figure]

140

**Figure S11. Time trends of selected AMS tracer ions in the ³C\*-aqSOA during aqSOA formation and prolonged aging.**

**Scheme S1. Postulated reaction pathways for the photodegradation of 3,4-dimethoxybenzaldehyde. The mechanisms are adapted from previous studies on benzaldehydes (Berger et al., 1973; Dubtsov et al., 2006; Shen and Fang, 2011; Theodoropoulou et al., 2020).**

---

## Editor Decision (ED1)

Editor comments

l. 44: Please clarify that these mass yields relate to the fraction of precursor present in the aqueous phase, not to the total phenol mass in the atmosphere.

l. 201/2: The references to the studies by Atkinson and Olariu are somewhat misleading here as they investigated phenol oxidation in the gas phase – which might not result in the same products and/or yields. Please make this clear in the text.

l. 233: What do you mean by 'in the meantime'? Simultaneously in the same step?

Improvements of figures:

Figures 1, 3 (and similar ones): Please use a different color scheme and/or vary symbol types for the various traces of the experiments (e.g. circles, squares, diamonds etc)
Currently, it is hard for readers with color vision deficiencies to distinguish the traces for the different experiments: https://www.color-blindness.com/coblis-color-blindness-simulator/

Figure 3: Please consider using a different color scheme other than 'rainbow scale' for the same reasons as above. Monochromatic schemes going from light to dark color shades are easier to distinguish.

Figures 6 and 7: What is the relevance or meaning of the fit equations (y = A(1-e^B x)) for the formation and decays? Can the rate constants are directly derived from these fit equations?
If so, please describe. If the coefficients are just empirical and are not further used, consider removing the equations from the quite cluttered figure and just refer to the tables in the supporting information.

 Figure 4: Please add the unit of the rate constant to the axis label.

Technical corrections

l. 352: Define 'AAE'

l. 388: reword 'than no extra oxidant' – do you mean 'than without extra oxidant'?

l. 394: Define 'OSc'

l. 431: Can you give an estimate of the relative contributions of aqSOA losses by chemical reactions vs (wet and dry) deposition? How much aqSOA is removed by deposition within 48 hours?

---

## Author Response (AR2)

We appreciate the editor's valuable comments, and we have revised the manuscript accordingly. Listed below are our point-to-point responses (in **blue**) to the comments (in **black**) and changes of the manuscript (in **red**).

**Editor comments**

l. 44: Please clarify that these mass yields relate to the fraction of precursor present in the aqueous phase, not to the total phenol mass in the atmosphere.

The text has been revised as follows for clarification.

The mass yields of aqSOA from the phenolic precursors in atmospheric waters range from 50% to 140%,

l. 201/2: The references to the studies by Atkinson and Olariu are somewhat misleading here as they investigated phenol oxidation in the gas phase – which might not result in the same products and/or yields. Please make this clear in the text.

To address this comment, we have revised the text to explicitly state that the studies by Atkinson and Olariu investigated phenol oxidation in the gas phase. Furthermore, to provide information on branching ratios and product distributions of phenol •OH oxidation in the aqueous phase, we have now included relevant modeling studies by Kılıç et al. in the references. The text has been updated as follows.

According to previous studies on phenol oxidation in the gas phase (Atkinson, 1986; Olariu et al., 2002), it has been observed that at room temperature, only ~10% of the phenol + •OH reaction involves H-atom abstraction that leads to the formation of phenoxy radical, whereas ~90% of the •OH reaction proceeds through OH addition. Moreover, modeling studies have indicated that in both gas-phase and aqueous-phase •OH oxidation of phenols, the OH addition pathways exhibit considerably lower activation energy than the H-abstraction pathway (Kılıç et al., 2007). As a result, it is highly likely that the primary products of the •OH reaction with phenols are hydroxyphenols.

l. 233: What do you mean by 'in the meantime'? Simultaneously in the same step?

Our intention was to convey that the H-atom abstraction/electron transfer between $^3$DMB* and GA results in the parallel formation of a GA phenoxyl radical and a DMB ketyl radical. To enhance clarity, the sentence has been revised as follows.

In $^3$C*-mediated reactions, $^3$C* can oxidize GA via H-atom abstraction/electron transfer to form a phenoxyl radial and/or a ketyl radical (Anastasio et al., 1997; Smith et al., 2014; Yu et al., 2014). The ketyl radical can react with $O_2$ to produce superoxide/hydroperoxyl radical ($O_2^{•-}/HO_2•$), which subsequently react to produce $H_2O_2$ (Anastasio et al., 1997).

**Improvements of figures:**

Figures 1, 3 (and similar ones): Please use a different color scheme and/or vary symbol types for the various traces of the experiments (e.g., circles, squares, diamonds etc). Currently, it is hard for readers with color vision deficiencies to distinguish the traces for the different experiments: https://www.color-blindness.com/coblis-color-blindness-simulator/

We have revised the color scheme and have incorporated different symbol types for the traces representing different experiments in Figures 1, 3, 6, 7, S10 and S11.

Figure 3: Please consider using a different color scheme other than 'rainbow scale' for the same reasons as above. Monochromatic schemes going from light to dark color shades are easier to distinguish.

We have revised the color scheme in Figures 3d-f and 3j-l by using lighter to darker shades instead of the rainbow scale.

Figures 6 and 7: What is the relevance or meaning of the fit equations (y = A(1-e^B x)) for the formation and decays? Can the rate constants be directly derived from these fit equations? If so, please describe. If the coefficients are just empirical and are not further used, consider removing the equations from the quite cluttered figure and just refer to the tables in the supporting information.

The fitted parameter B represents the first-order formation rate constant of an aqSOA factor in the photoreactor under the experiment conditions. The following sentences have been added to Section 3.3 for clarification.

The formation and decay rate constants of different generations of the aqSOA products were determined by performing exponential fits ($y = a(1-e^{-bx}) + c$ and $y = ae^{-bx} + c$, respectively) to the time trends of the aqSOA factors (Figures 6d,f,h and 7d,f,h). The fitted parameter b (in the unit of $h^{-1}$) represents the first-order rate constant for the aqSOA formation or decay in the photoreactor.

Figure 4: Please add the unit of the rate constant to the axis label.

Corrected.

**Technical corrections**

l. 352: Define 'AAE'

Updated.

l. 388: reword 'than no extra oxidant' – do you mean 'than without extra oxidant'?

Yes. The text has been revised to 'than without extra oxidant'.

l. 394: Define 'OSc'

The definition of OSc has been provided in Section 2.2 (Line 139-140) of the manuscript. To address this comment, we have now reiterated the definition of OSc in Line 394.

l. 431: Can you give an estimate of the relative contributions of aqSOA losses by chemical reactions vs (wet and dry) deposition? How much aqSOA is removed by deposition within 48 hours?

In this study, the rate of loss for phenolic aqSOA during photochemical aging was found to be in the range of 0.017–0.11 h$^{-1}$ (i.e., 5–30×10$^{-6}$ s$^{-1}$). The photochemical kinetics in our RPR-200 photoreactor system were ~7 times faster than those experienced under ambient winter solstice sunlight in Northern California. Consequently, these observations indicate a photochemical lifetime of 3–17 days for phenolic aqSOA in ambient conditions. The deposition loss rate constant of submicron particles in the atmosphere, assuming wet deposition is the dominant process, is approximately $2×10^{-6}$ s$^{-1}$ (resulting in a lifetime of approximately 5 days) (Henry and Donahue, 2012; Molina et al., 2004). These findings suggest that the contribution of photochemical aging to the removal of phenolic aqSOA can be comparable to that of wet deposition.

We have included this discussion in the conclusion section of the updated manuscript.

**Reference**

Anastasio, C., Faust, B. C. and Rao, C. J.: Aromatic Carbonyl Compounds as Aqueous-Phase Photochemical Sources of Hydrogen Peroxide in Acidic Sulfate Aerosols, Fogs, and Clouds. 1. Non-Phenolic Methoxybenzaldehydes and Methoxyacetophenones with Reductants (Phenols), Environ. Sci. Technol., 31(1), 218–232, doi:10.1021/es960359g, 1997.

Atkinson, R.: Kinetics and mechanisms of the gas-phase reactions of the hydroxyl radical with organic compounds under atmospheric conditions, Chem. Rev., 86(1), 69–201, doi:10.1021/cr00071a004, 1986.

Henry, K. M. and Donahue, N. M.: Photochemical Aging of α-Pinene Secondary Organic Aerosol: Effects of OH Radical Sources and Photolysis, J. Phys. Chem. A, 116(24), 5932–5940, doi:10.1021/jp210288s, 2012.

Kılıç, M., Koçtürk, G., San, N. and Çınar, Z.: A model for prediction of product distributions for the reactions of phenol derivatives with hydroxyl radicals, Chemosphere, 69(9), 1396–1408, doi:https://doi.org/10.1016/j.chemosphere.2007.05.002, 2007.

Molina, M. J., Ivanov, A. V, Trakhtenberg, S. and Molina, L. T.: Atmospheric evolution of organic aerosol, Geophys. Res. Lett., 31(22), doi:https://doi.org/10.1029/2004GL020910, 2004.

Olariu, R. I., Klotz, B., Barnes, I., Becker, K. H. and Mocanu, R.: FT–IR study of the ring-retaining products from the reaction of OH radicals with phenol, o-, m-, and p-cresol, Atmos. Environ., 36(22), 3685–3697, doi:https://doi.org/10.1016/S1352-2310(02)00202-9, 2002.

Smith, J. D., Sio, V., Yu, L., Zhang, Q. and Anastasio, C.: Secondary Organic Aerosol Production

from Aqueous Reactions of Atmospheric Phenols with an Organic Triplet Excited State, Environ. Sci. Technol., 48(2), 1049–1057, doi:10.1021/es4045715, 2014.

Yu, L., Smith, J., Laskin, A., Anastasio, C., Laskin, J. and Zhang, Q.: Chemical characterization of SOA formed from aqueous-phase reactions of phenols with the triplet excited state of carbonyl and hydroxyl radical, Atmos. Chem. Phys., 14(24), 13801–13816, doi:10.5194/acp-14-13801-2014, 2014.